# A Trust Crisis In Simulation-Based Inference? Your Posterior Approximations Can Be Unfaithful

**Joeri Hermans**[*]                                                     *joeri@peinser.com*
*Unaffiliated*

**Arnaud Delaunoy**[*]                                              *a.delaunoy@uliege.be*
*University of Liège*

**François Rozet**                                              *francois.rozet@uliege.be*
*University of Liège*

**Antoine Wehenkel**                                        *antoine.wehenkel@uliege.be*
*University of Liège*

**Volodimir Begy**                                          *volodimir.begy@univie.ac.at*
*University of Vienna*

**Gilles Louppe**                                                     *g.louppe@uliege.be*
*University of Liège*

**Reviewed on OpenReview:** *https://openreview.net/forum?id=LHAbHkt6Aq*

## Abstract

We present extensive empirical evidence showing that current Bayesian simulation-based inference algorithms can produce computationally unfaithful posterior approximations. Our results show that all benchmarked algorithms – (Sequential) Neural Posterior Estimation, (Sequential) Neural Ratio Estimation, Sequential Neural Likelihood and variants of Approximate Bayesian Computation – can yield overconfident posterior approximations, which makes them unreliable for scientific use cases and falsificationist inquiry. Failing to address this issue may reduce the range of applicability of simulation-based inference. For this reason, we argue that research efforts should be made towards theoretical and methodological developments of conservative approximate inference algorithms and present research directions towards this objective. In this regard, we show empirical evidence that ensembling posterior surrogates provides more reliable approximations and mitigates the issue.

## 1 Introduction

Many scientific disciplines rely on computer simulations to study complex phenomena under various conditions. Although modern simulators can generate realistic synthetic observables through detailed descriptions of their data generating processes, they make statistical inference more challenging. The computer code describing the data generating processes defines the likelihood function $p(\boldsymbol{x}|\boldsymbol{\theta})$ only implicitly, and its direct evaluation requires the often intractable integration of all stochastic execution paths. In this problem setting, statistical inference based on the likelihood becomes impractical. However, approximate inference remains possible by relying on likelihood-free approximations thanks to the increasingly accessible and effective suite of methods and software from the field of simulation-based inference (Cranmer et al., 2020).

While simulation-based inference targets domain sciences, advances in the field are mainly driven from a machine learning perspective. The field, therefore, inherits the quality assessments (Lueckmann et al.,

---

[*]Equal contribution

2021) customary to the machine learning literature, primarily targeting the exactness of the approximation. In fact, domain sciences, and more specifically the physical sciences, are not necessarily interested in the exactness of an approximation. Instead, in the tradition of Popperian falsification, they often seek to **constrain parameters** of interest as much as possible at a given confidence level. Scientific examples include frequentist confidence intervals on the mass of the Higgs boson (Aad et al., 2012), Bayesian credible regions on cosmological parameters (Gilman et al., 2018; Aghanim et al., 2020), constraints on the intrinsic parameters of binary black hole coalescences (Abbott et al., 2016), or characterizing the space of circuit configurations giving rise to rhythmic activity in the crustacean stomatogastric ganglion (Gonçalves et al., 2020). Wrongly excluding plausible values could drive the scientific inquiry in the wrong direction, whereas failing to exclude implausible values because of too conservative estimations is much less detrimental. This implies that statistical approximations in simulation-based inference should ideally come with conservative guarantees to not produce credible regions smaller than they should be, even when the approximations are not faithful. Despite recent developments of post hoc diagnostics to inspect the quality of likelihood-free (Cranmer et al., 2015; Brehmer et al., 2018; 2019; Hermans et al., 2021; Lueckmann et al., 2021; Talts et al., 2018; Dalmasso et al., 2020) and likelihood-based (Geweke et al., 1991; Gelman & Rubin, 1992; Raftery & Lewis, 1991; Dixit & Roy, 2017; Talts et al., 2018) approximations, assessing whether approximate inference results are sufficiently reliable for scientific inquiry remains largely unanswered whenever fitting criteria are not globally optimized or whenever the data is limited.

In this work, we measure and discuss the quality of the credible regions produced by various algorithms for Bayesian simulation-based inference. We frame our main contribution as the collection of extensive empirical evidence requiring months of computation, demonstrating that all benchmarked techniques may produce non-conservative credible regions. Our results emphasize the need for a new class of methods: conservative approximate inference algorithms. In addition, we provide empirical evidence that using an ensemble of models in place of a single model tends to produce more reliable approximations. The structure of the paper is outlined as follows. Section 2 describes the statistical formalism, necessary background and includes a thorough motivation for coverage. Section 3 highlights our main results. Section 4 presents several avenues of future research to enable drawing reliable scientific conclusions with simulation-based inference. All code related to this manuscript is available at `https://github.com/montefiore-ai/trust-crisis-in-simulation-based-inference`.

## 2 Background

### 2.1 Statistical formalism

We evaluate posterior estimators that produce approximations $\hat{p}(\boldsymbol{\theta}|\boldsymbol{x})$ with the following semantics.

**Target parameters $\boldsymbol{\theta}$** denote the parameters of interest of a simulation model, and are sometimes referred to as free or model parameters. We make the reasonable assumption that the prior $p(\boldsymbol{\theta})$ is tractable.

An **observable $\boldsymbol{x}$** denotes a synthetic realization of the simulator, or the observed data $\boldsymbol{x}_o$ we would like to do inference on. We assume that the simulation model is correctly specified and hence is an accurate representation of the real data generation process.

The **likelihood** model $p(\boldsymbol{x}|\boldsymbol{\theta})$ is implicitly defined by the simulator's computer code. While we cannot evaluate the density $p(\boldsymbol{x}|\boldsymbol{\theta})$, we can draw samples through simulation.

The **ground truth $\boldsymbol{\theta}^*$** specified to the simulation model whose forward evaluation produced the observable $\boldsymbol{x}_o$, i.e., $\boldsymbol{x}_o \sim p(\boldsymbol{x}|\boldsymbol{\theta} = \boldsymbol{\theta}^*)$.

A **credible region** is a space $\Theta$ within the target parameters domain that satisfies

$$\int_{\Theta} p(\boldsymbol{\theta}|\boldsymbol{x} = \boldsymbol{x}_o)\,\mathrm{d}\boldsymbol{\theta} = 1 - \alpha \tag{1}$$

for some observable $\boldsymbol{x}_o$ and confidence level $1 - \alpha$. Because many such regions exist, we compute the credible region with the smallest volume. In the literature this credible region is known as the highest posterior density region (Box & Tiao, 1973; Hyndman, 1996). In our evaluations, we determine the credible regions by evaluating the approximated posterior density function in a discretized and empirically normalized grid

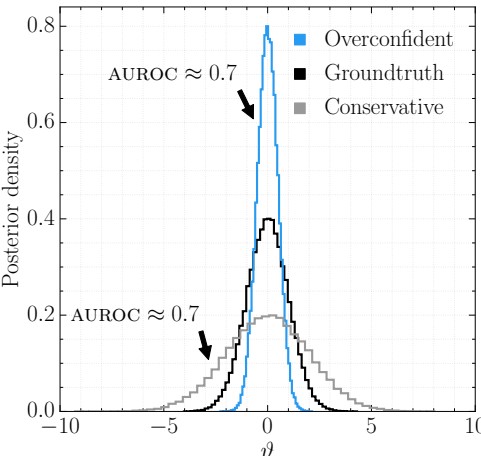

Figure 1: A classifier-based metric measures the divergence between posterior approximations and a ground truth by means of evaluating the classifier's discriminative performance through Area Under the Receiver Operating Characteristics curve (AUROC). The metric indicates that both the overconfident and the conservative approximations are equally accurate as it yields AUROC = 0.7 for both of them. From an inference perspective however, the conservative approximation is more suitable because it produces credible regions larger than they should be.

of the parameter space. The credible region is the set of parameters whose density is greater than a given threshold fitted to achieve the desired credibility level $1 - \alpha$. Additional details are described in Appendix A.

## 2.2 Statistical quality assessment

Common metrics for evaluating the quality of a posterior surrogate include the Classifier Two-sample Test (Lehmann & Romano, 2006; Lopez-Paz & Oquab, 2017) and Maximum Mean Discrepancy (Gretton et al., 2012; Bengio et al., 2014; Dziugaite et al., 2015). The main problem with these metrics is that they assess exactness of an approximation through a divergence with respect to the posterior. All approximations will diverge from the posterior and there are no criteria to what constitutes an acceptable estimator. For this reason, we argue that metrics evaluating the reliability for scientific inquiry should be used alongside the divergence evaluation when evaluating estimators.

To clarify this point, consider the demonstration in Figure 1. A binary classifier is trained to discriminate between samples from a posterior approximation and the true posterior, as in a classifier two-sample test. The discriminative performance of the classifier is expressed through Area Under the Receiver Operating Characteristics curve (AUROC) and serves as a measure for divergence between both densities. An AUROC = 0.5 suggests an approximation that is indistinguishable from the true posterior, while AUROC = 1.0 implies that both distributions do not overlap. Although both the overconfident and the conservative approximations are of equal quality according to the AUROC metric (AUROC = 0.7), credible regions that are biased or smaller than they should be could result in the wrong exclusion of actually plausible parameter values for a given significance level, and hence to erroneous scientific conclusions. By contrast, conservative approximations, which leads to credible regions that are larger than they should be, are more scientifically reliable since they would not wrongly reject plausible parameters values but only fail to reject actually implausible parameter values. For this reason, we take the position that posterior approximations should produce overdispersed credible regions for any simulation budget. Posterior approximations do not have to closely match the true posterior to draw meaningful inferences, but they should however be conservative.

Instead of measuring the exactness of an approximation, this work directly assesses the quality of credible regions through the notion of expected coverage which probes the consistency of the posterior approximations and can be used to diagnose conservative and overconfident approximations. Similar usages of coverage

diagnostics can be found in the context of standard statistical inference (Schall, 2012) and approximate Bayesian computation (Prangle et al., 2014).

**Definition 2.1.** The **expected coverage probability** of the $1 - \alpha$ highest posterior density regions derived from the posterior estimator $\hat{p}(\boldsymbol{\theta}|\boldsymbol{x})$ is

$$\mathbb{E}_{p(\boldsymbol{\theta},\boldsymbol{x})}\left[\mathbb{1}\left(\boldsymbol{\theta} \in \Theta_{\hat{p}(\boldsymbol{\theta}|\boldsymbol{x})}(1-\alpha)\right)\right], \tag{2}$$

where the function $\Theta_{\hat{p}(\boldsymbol{\theta}|\boldsymbol{x})}(1-\alpha)$ yields the $1-\alpha$ highest posterior density region of $\hat{p}(\boldsymbol{\theta}|\boldsymbol{x})$.

Note that Equation 2 can be expressed either as

$$\mathbb{E}_{p(\boldsymbol{\theta})}\mathbb{E}_{p(\boldsymbol{x}|\boldsymbol{\theta})}\left[\mathbb{1}\left(\boldsymbol{\theta} \in \Theta_{\hat{p}(\boldsymbol{\theta}|\boldsymbol{x})}(1-\alpha)\right)\right], \tag{3}$$

which is the expected frequentist coverage probability, or alternatively as the expected Bayesian credibility

$$\mathbb{E}_{p(\boldsymbol{x})}\mathbb{E}_{p(\boldsymbol{\theta}|\boldsymbol{x})}\left[\mathbb{1}\left(\boldsymbol{\theta} \in \Theta_{\hat{p}(\boldsymbol{\theta}|\boldsymbol{x})}(1-\alpha)\right)\right], \tag{4}$$

whose inner expectation reduces to $1 - \alpha$ whenever the posterior estimator $\hat{p}(\boldsymbol{\theta}|\boldsymbol{x})$ is well-calibrated.

The expected coverage probability can be estimated empirically given a set of $n$ i.i.d. samples $(\boldsymbol{\theta}_i^*, \boldsymbol{x}_i) \sim p(\boldsymbol{\theta}, \boldsymbol{x})$ as

$$\frac{1}{n}\sum_{i=1}^{n}\mathbb{1}\left(\boldsymbol{\theta}_i^* \in \Theta_{\hat{p}(\boldsymbol{\theta}|\boldsymbol{x}_i)}(1-\alpha)\right). \tag{5}$$

**Definition 2.2.** A **conservative posterior estimator** is an estimator that **has coverage** at the credibility level of interest, i.e., whenever the expected coverage probability is larger or equal to the credibility level.

While coverage is necessary to assess conservativeness, it is limited in its ability to determine the information gain a posterior (approximation) has over its prior. Consider an estimator whose posteriors are identical to the prior. In this case, there is no gain in information and the expected coverage probability is equal to the credibility level. For this reason, a complete analysis should be complemented with measures such as the expected information gain $\mathbb{E}_{p(\boldsymbol{\theta},\boldsymbol{x})}\left[\log p(\boldsymbol{\theta}|\boldsymbol{x}) - \log p(\boldsymbol{\theta})\right]$ (as illustrated in Appendix D) or classifier two-sample tests when the ground-truth posterior is available. This work is concerned with **conservative inference** and will therefore mainly focus on the evaluation of expected coverage. Finally, it should be noted that expected coverage is a statement about the credible regions in expectation and therefore does not provide any guarantee on the quality of a single posterior approximation. However, the quality of a single posterior approximation can itself be approximated with a local coverage test (Zhao et al., 2021).

## 3 Empirical observations

This section covers our main contribution: the collection of empirical evidence to determine whether some simulation-based inference algorithms are conservative by nature. We are particularly interested in determining whether certain approaches should be favoured over others. We do so by estimating the expected coverage of posterior estimators produced by these approaches across a broad range of hyperparameters and benchmarks of varying complexity, including two real problems. As in real use cases, the true posteriors are effectively intractable and therefore unknown.

### 3.1 Methods

We make the distinction between two paradigms. *Non-amortized* approaches are designed to approximate a single posterior, while *amortized* methods aim to learn a general purpose estimator that attempts to approximate all posteriors supported by the prior. The architectures used for each inference algorithm, including hyperparameters, are listed in Appendix C.

### 3.1.1 Amortized

**Neural Ratio Estimation** (NRE) is an established approach in the simulation-based inference literature both from a frequentist (Cranmer et al., 2015) and Bayesian (Thomas et al., 2016; Hermans et al., 2020; Durkan et al., 2020) perspective. In a Bayesian analysis, an amortized estimator $\hat{r}(\boldsymbol{x}|\boldsymbol{\theta})$ of the intractable likelihood-to-evidence ratio $r(\boldsymbol{x}|\boldsymbol{\theta})$ can be learned by training a binary classifier $\hat{d}(\boldsymbol{\theta}, \boldsymbol{x})$ to distinguish between samples of the joint $p(\boldsymbol{\theta}, \boldsymbol{x})$ with class label 1 and samples of the product of marginals $p(\boldsymbol{\theta})p(\boldsymbol{x})$ with class label 0, with equal label marginal probability. Similar to the density-ratio trick (Sugiyama et al., 2012; Goodfellow et al., 2014; Cranmer et al., 2015; Hermans et al., 2020), the Bayes optimal classifier $d(\boldsymbol{\theta}, \boldsymbol{x})$ for the cross-entropy loss is

$$d(\boldsymbol{\theta}, \boldsymbol{x}) = \frac{p(\boldsymbol{\theta}, \boldsymbol{x})}{p(\boldsymbol{\theta}, \boldsymbol{x}) + p(\boldsymbol{\theta})p(\boldsymbol{x})} = \sigma\left(\log \frac{p(\boldsymbol{\theta}, \boldsymbol{x})}{p(\boldsymbol{\theta})p(\boldsymbol{x})}\right), \tag{6}$$

where $\sigma(\cdot)$ is the sigmoid function. Given a target parameter $\boldsymbol{\theta}$ and an observable $\boldsymbol{x}$ supported by $p(\boldsymbol{\theta})$ and $p(\boldsymbol{x})$ respectively, the learned classifier $\hat{d}(\boldsymbol{\theta}, \boldsymbol{x})$ approximates the log likelihood-to-evidence ratio $\log r(\boldsymbol{x}|\boldsymbol{\theta})$ through the logit function because $\text{logit}(\hat{d}(\boldsymbol{\theta}, \boldsymbol{x})) \approx \log r(\boldsymbol{x}|\boldsymbol{\theta})$. The approximate log posterior density function is $\log p(\boldsymbol{\theta}) + \log \hat{r}(\boldsymbol{x}|\boldsymbol{\theta})$.

**Neural Posterior Estimation** (NPE) (Rezende & Mohamed, 2015) is concerned with directly learning an amortized posterior estimator $\hat{p}_\psi(\boldsymbol{\theta}|\boldsymbol{x})$ with normalizing flows. Normalizing flows define a class of probability distributions $p_\psi(\cdot)$ built from neural network-based bijective transformations (Rezende & Mohamed, 2015; Dinh et al., 2015; 2017) parameterized by $\psi$. They are usually optimized using variational inference, by solving $\arg\min_\psi \mathbb{E}_{p(\boldsymbol{x})}\left[\text{KL}(p(\boldsymbol{\theta}|\boldsymbol{x}) \,||\, \hat{p}_\psi(\boldsymbol{\theta}|\boldsymbol{x})\right]$, which is equivalent to $\arg\max_\psi \mathbb{E}_{p(\boldsymbol{\theta}, \boldsymbol{x})}\left[\log \hat{p}_\psi(\boldsymbol{\theta}|\boldsymbol{x})\right]$. Once trained, the density of the modeled distribution can directly be evaluated and sampled from.

**Ensembles** of models constitute a standard method to improve predictive performance. In this work, we consider an ensemble model that averages the approximated posteriors of $n$ posterior estimators that are either trained independently on the same dataset (deep ensemble) Lakshminarayanan et al. (2017) or on bootstrap replicates of the learning set (bagging) Breiman (1996). While this formulation is natural for NPE, averaging likelihood-to-evidence ratios is equivalent since $\frac{1}{n}\sum_{i=1}^n \hat{p}_i(\boldsymbol{\theta}|\boldsymbol{x}) = p(\boldsymbol{\theta})\frac{1}{n}\sum_{i=1}^n \hat{r}_i(\boldsymbol{x}|\boldsymbol{\theta})$. We show in Section 3 that ensembles lead in average to a higher expected coverage than individual models and hence constitute a straightforward mitigation strategy against overconfidence.

### 3.1.2 Non-amortized

**Rejection Approximate Bayesian Computation** (REJ-ABC) (Rubin, 1984; Pritchard et al., 1999) numerically estimates a single posterior by collecting samples $\boldsymbol{\theta} \sim p(\boldsymbol{\theta})$ whenever $\boldsymbol{x} \sim p(\boldsymbol{x}|\boldsymbol{\theta})$ is similar to $\boldsymbol{x}_o$. Similarity is expressed by means of a distance function $\rho$. For high-dimensional observables, the probability density of simulating an observable $\boldsymbol{x}$ such that $\boldsymbol{x} = \boldsymbol{x}_o$ is extremely small. For this reason, ABC uses a summary statistic $s$ and an acceptance threshold $\epsilon$. Using these components, ABC accepts samples into the approximate posterior whenever $\rho(s(\boldsymbol{x}), s(\boldsymbol{x}_o)) \leq \epsilon$. Improvements include regression adjustment (Beaumont et al., 2002) of the sampled parameters using local linear regression, combining ABC with MCMC (Marjoram et al., 2003) and the automatic construction of summary statistics (Fearnhead & Prangle, 2012). In our experiments, we apply REJ-ABC in its simplest form and use the identity function as a sufficient summary statistic, use no regression adjustment and set $\epsilon$ such that $\max(100, \text{simulation budget}/100)$ samples are accepted.

Sequential methods for simulation-based inference aim to approximate a single posterior by iteratively improving a posterior approximation. These methods alternate between a simulation and an exploitation phase. The latter is designed to take current knowledge into account such that subsequent simulations can be focused on parameters that are more likely to produce observables $x$ similar to $x_o$.

**Sequential Monte-Carlo ABC** (SMC-ABC) (Toni & Stumpf, 2009; Sisson et al., 2007; Beaumont et al., 2009) iteratively updates a set of proposal states to match the posterior distribution. At each iteration, accepted proposals are ranked by distance. The rankings determine whether a proposal is propagated to the next iteration. New candidates are generated by perturbing the selected ranked proposals.

**Sequential Neural Posterior Estimation** (SNPE) (Papamakarios & Murray, 2016; Lueckmann et al., 2017; Greenberg et al., 2019) iteratively improves a normalizing flow that models the posterior. Our evaluations will specifically use the SNPE-C (Greenberg et al., 2019) variant.

**Sequential Neural Likelihood** (SNL) (Papamakarios et al., 2019) models the likelihood $p(\boldsymbol{x}|\boldsymbol{\theta})$. A numerical approximation of the posterior is obtained by plugging the learned likelihood estimator into a Markov Chain Monte Carlo (MCMC) sampler as a surrogate likelihood. Similarly, Price et al. (2018); Frazier et al. (2022) construct a synthetic normal approximation of the likelihood over summary statistics.

**Sequential Neural Ratio Estimation** (SNRE) (Hermans et al., 2020; Durkan et al., 2020) iteratively improves the modelled likelihood-to-evidence ratio.

### 3.2 Benchmarks

We consider 7 benchmarks, ranging from a toy problem to real scientific use cases covering various disciplines. All benchmarks and priors are available in the codebase.

The **SLCP** simulator models a fictive problem with 5 parameters. The observable $\boldsymbol{x} \in \mathbb{R}^8$ represents the 2D-coordinates of 4 points. The coordinate of each point is sampled from the same multivariate Gaussian whose mean and covariance matrix are parametrized by $\boldsymbol{\theta}$. We consider an alternative version of the original task (Papamakarios et al., 2019) by inferring the marginal posterior density of 2 of those parameters. In contrast to its original formulation, the likelihood is not tractable due to the marginalization.

The **Weinberg** problem (Cranmer et al., 2017) concerns a simulation of high energy particle collisions $e^+e^- \to \mu^+\mu^-$. The angular distribution of the particles can be used to measure the Weinberg angle $\boldsymbol{x}$ in the standard model of particle physics. From the scattering angle, we are interested in inferring Fermi's constant $\boldsymbol{\theta}$.

The **Spatial SIR** model involves a grid-world of susceptible, infected, and recovered individuals. Based on initial conditions and the infection and recovery rate $\boldsymbol{\theta}$, the model describes the spatial evolution of an infection. The observable $\boldsymbol{x}$ is a snapshot of the grid-world after some fixed amount of time.

**M/G/1** (Blum & François, 2010) models a processing and arrival queue. The problem is described by 3 parameters $\boldsymbol{\theta}$ that influence the time it takes to serve a customer, and the time between their arrivals. The observable $\boldsymbol{x}$ is composed of 5 equally spaced quantiles of inter-departure times.

The **Lotka-Volterra** population model (Lotka, 1920; Volterra, 1926) describes a process of interactions between a predator and a prey species. The model is conditioned on 4 parameters $\boldsymbol{\theta}$ which influence the reproduction and mortality rate of the predators and preys. We infer the marginal posterior of the predator parameters from time series representing the evolution of both populations over time.

Stellar **Streams** form due to the disruption of spherically packed clusters of stars by the Milky Way. Because of their distance from the galactic center and other visible matter, distant stellar streams are considered to be ideal probes to detect gravitational interactions with dark matter. The model (Banik et al., 2018) evolves the stellar density $\boldsymbol{x}$ of a stream over several billion years and perturbs the stream over its evolution through gravitational interactions with dark matter subhaloes parameterized by the dark matter mass $\boldsymbol{\theta}$.

**Gravitational Waves (GW)** are ripples in space-time emitted during events such as the collision of two black-holes. They can be detected through interferometry measurements $\boldsymbol{x}$ and convey information about celestial bodies, unlocking new ways to study the universe. We consider inferring the masses $\boldsymbol{\theta}$ of two black-holes colliding through the observation of the gravitational wave as measured by LIGO's dual detectors (LIGO Scientific Collaboration, 2018; Biwer et al., 2019).

### 3.3 Setup

Our evaluations consider simulation budgets ranging from $2^{10}$ up to $2^{17}$ samples and confidence levels from 0.05 up to 0.95. Within the amortized setting we train, for every simulation budget, 5 posterior estimators for 100 epochs. The expected coverage probability is estimated as described in Appendix A on at least 5,000 unseen samples from the joint $p(\boldsymbol{\theta}, \boldsymbol{x})$ and for all confidence levels under consideration. In addition, we

repeat the expected coverage evaluation for ensembles of 5 estimators as well. Special care for non-amortized approaches is necessary because they approximate a single posterior and can therefore not reasonably evaluate expected coverage in the same way. Our experiments for non-amortized approaches estimate the expected coverage by repeating the inference procedure on 300 distinct observables for every simulation budget. The expected coverage probabilities are subsequently estimated based on the resulting posterior approximations. Our experiments with NPE, SNPE, SNL, SNRE, REJ-ABC and SMC-ABC rely on the implementation in the `sbi` package (Tejero-Cantero et al., 2020), while a custom implementation for NRE is used.

**Computational cost** We emphasize the computational requirements necessary to generate our main contribution: the experimental observations, whose generation took months of computation. The average CPU time for evaluating an amortized procedure on all common benchmark problems is in order of 200 CPU days, while for a non-amortized approach this increases to 2800 CPU days. The bulk of the cost was associated with the repeated optimization procedure and execution of the simulator. While we considered benchmarks with low dimensional parameter spaces (up to 3 parameters), we stress that the cost for computing credible regions would grow exponentially with the number of parameters, making coverage diagnostics expensive for high-dimensional problems. Note that when using NPE, the procedure for computing the coverages could be modified to scale to higher dimensions (Rozet & Louppe, 2021a;b). However, this procedure cannot be applied to all simulation-based inference algorithms. Nevertheless, we expect the observed behaviours to be similar or accentuated in a high-dimensional setting.

### 3.4 Results

Figures 2 and 3 highlight our main results. Through these plots, we can directly assess whether a posterior estimator is conservative at a given confidence level and simulation budget. The figures should be interpreted as follows: a perfectly calibrated posterior has an expected coverage probability equal to the credibility level. Plotting this relation produces a diagonal line. Conservative estimators on the other hand produce curves above the diagonal and overconfident models underneath. The plots highlight an unsettling observation: **all benchmarked approaches produce non-conservative posterior approximations on at least one problem setting**. In general, this pathology is especially prominent in non-amortized approaches with a small simulation budget; a regime they have been specifically designed for. A large simulation budget does not guarantee that a posterior estimator is conservative either. More importantly, as the reliability of the approximations computed through sequential approaches cannot practically be determined, it leaves practitioners uncertain about the reliability of their approximations. To complement this analysis, an evaluation of the predictive performance of the approximate posteriors is provided in Appendix D. We observe that, on most benchmark/method pairs, the predictive performance increases with the simulation budget. However, this is not the case for SNRE applied to MG1 which explains the strange behaviour of the coverage curves.

In sequential approaches, overconfidence could be explained by the alternating simulation and exploitation phases. One potential failure mode is that a non-conservative posterior approximation at a previous iteration forces the subsequent simulation phases to not produce observables that should in fact be associated with a higher posterior density, causing the posterior estimator to increase its overconfidence at each iteration.

Despite the fact that all ABC approaches use a sufficient summary statistic (by definition, the identity function), our results demonstrate that this alone is no guarantee for conservative posterior approximations. In fact, using a sufficient summary statistic with $\epsilon > 0$ does not always correspond to conservative approximations either. In such cases, ABC accepts samples with larger distances, permitting the procedure to shift the mass of the approximated posterior elsewhere. In addition, a limited number of posterior samples can negatively affect the quality of the credible regions, e.g., when approximating the posterior density function with kernel density estimation. Both cases can cause the observed behaviour. ABC should therefore be applied with caution to scientific applications. Even though a handcrafted, albeit sufficient, summary statistic provides some insight into the approximated posterior, it does not imply that ABC approximations are conservative whenever the threshold $\epsilon > 0$.

In Figure 3, we observe that the expected coverage probability of deep ensemble models is consistently larger than the expected coverage probability of an individual posterior estimator. **This highlights the fact that**

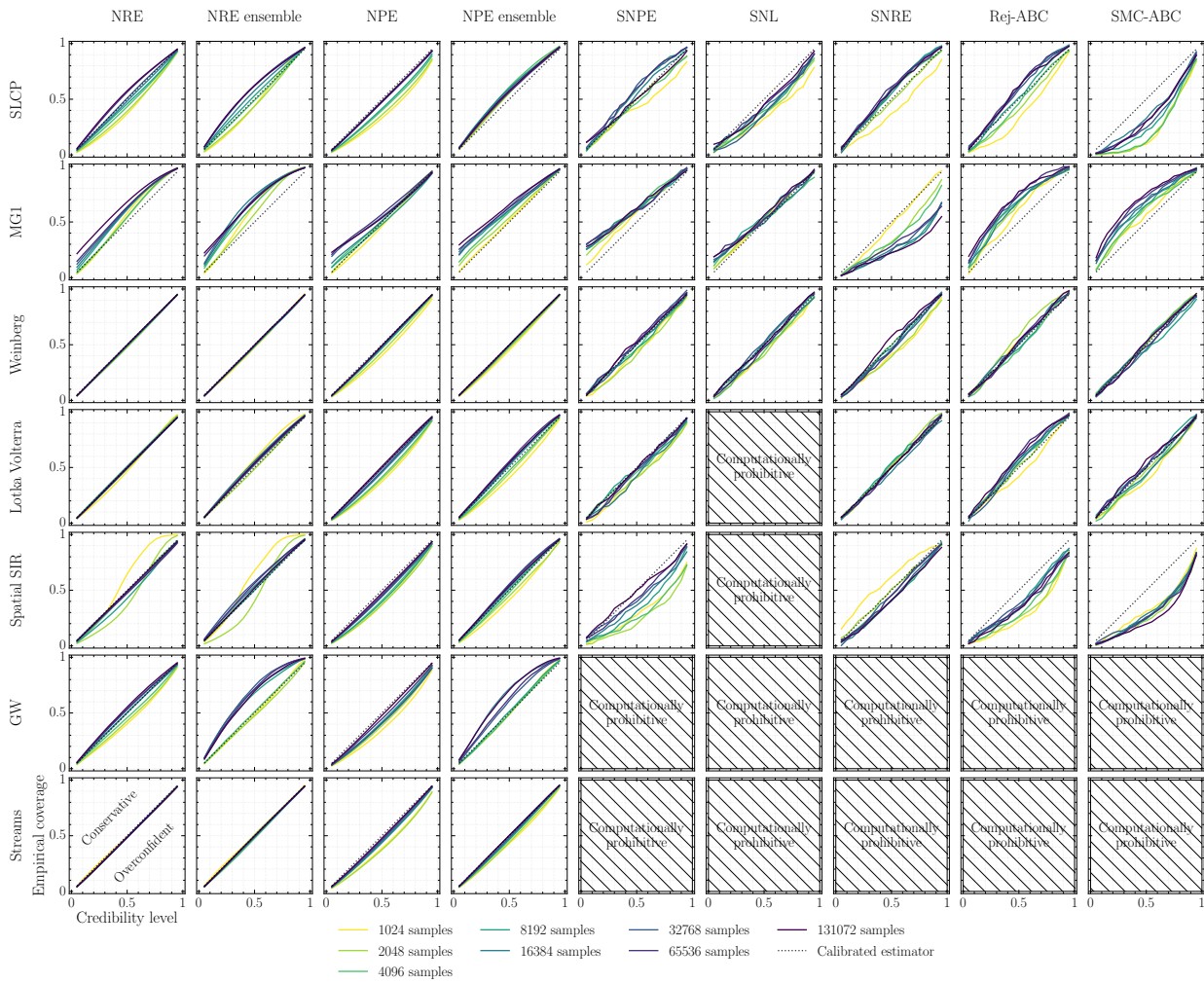

Figure 2: Evolution of the expected coverage w.r.t the simulation budget. A perfectly calibrated posterior has an expected coverage probability equal to the credibility level and produces a diagonal line. Conservative estimators on the other hand produce curves above the diagonal and overconfident models underneath. All algorithms can lead to non-conservative estimators. This pathology tends to be accentuated for small simulation budgets and non-amortized methods. Finally, the computationally prohibitive results indicate that the computational requirements did not allow for a coverage analysis. In the case of SNL, this was mostly due to high dimensional observables. For the astronomy benchmarks, the simulation model was simply too expensive to reasonably evaluate coverage for non-amortized methods.

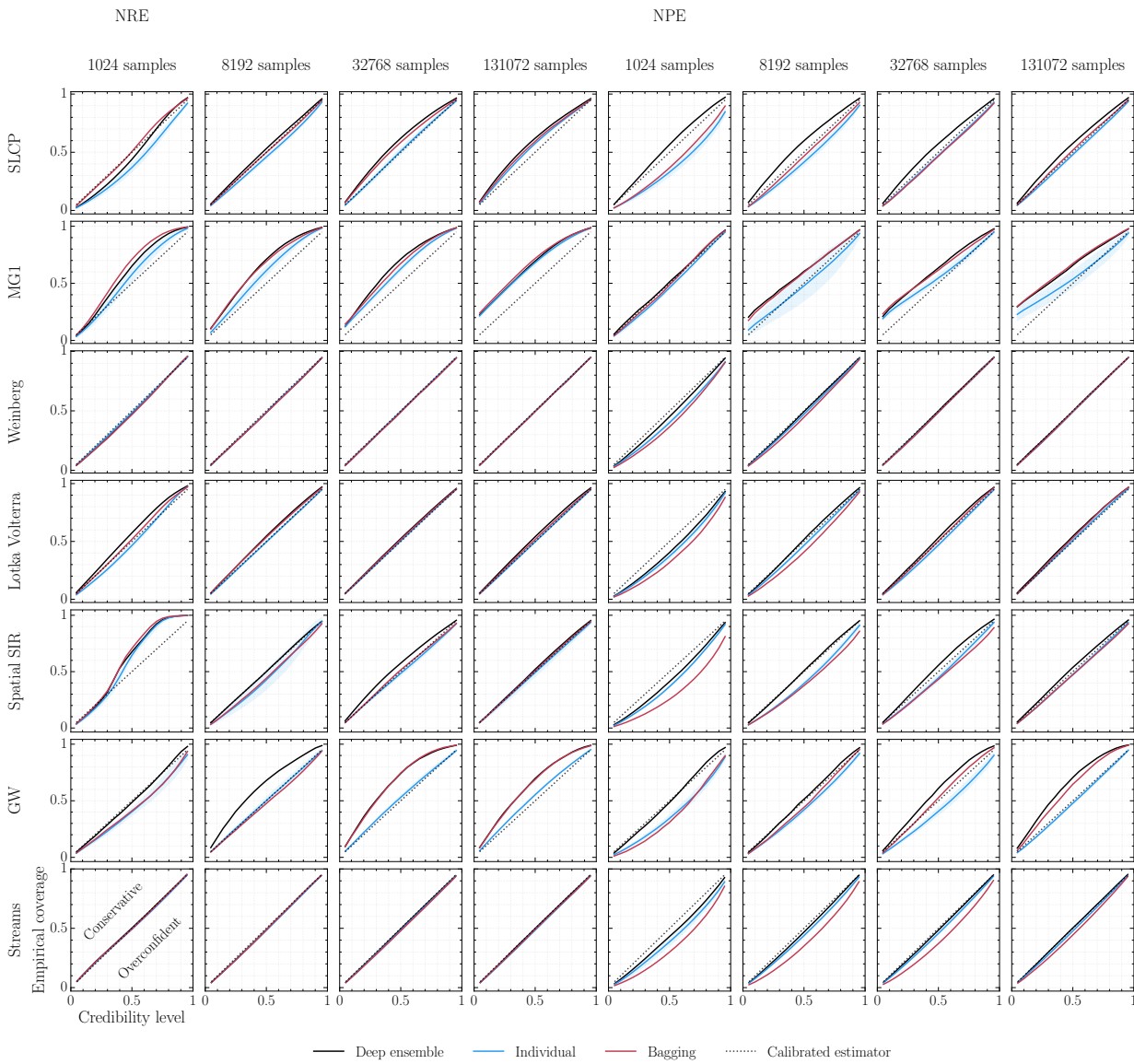

Figure 3: Analysis of coverage between ensemble and individual models w.r.t the various simulation budgets. The blue line represents the mean expected coverage of individual models over 5 runs, the shaded area represents its standard deviation. The black line represents the expected coverage of a single ensemble composed of 5 models. We observe that deep ensembles consistently have a higher expected coverage probability compared to the average individual model. A similar effect is not always observed with bagging, indicated by the red line. Ensembles are only evaluated for amortized approaches such as NPE and NRE.

**ensembling constitutes an immediately applicable and easy way to mitigate the overconfidence issue and build more reliable posterior estimators**. However, the deep ensemble model can still be non-conservative. We hypothesize that the increase in coverage is linked to the added uncertainty captured by the ensemble model, leading to inflated credible regions. In fact, individual estimators only capture data uncertainty, while an ensemble is expected to partially capture the epistemic uncertainty as well. Surprisingly, we find that ensembles built using bagging do not always produce higher coverage than individual models while they should also capture part of the epistemic uncertainty. We could potentially attribute this behaviour to the reduced effective dataset size used to train each member of the ensemble. Appendix E shows a positive effect with respect to ensemble size.

Not evident from figures 2 and 3 are the computational consequences of a coverage analysis on non-amortized methods. Although the figures mention a certain simulation budget, the total number of simulations for non-amortized methods should be multiplied by the number of approximated posteriors (300) to estimate the coverage. This highlights the simulation cost associated with diagnosing non-amortized approaches. This issue is not limited to coverage. Simulation-Based Calibration (SBC) (Talts et al., 2018) relies on samples of arbitrary posterior approximations as well. Diagnosing non-amortized estimators with SBC therefore requires a similar approach as we have taken in our coverage analyses. In fact, other benchmark papers such as Lueckmann et al. (2021) restrain from evaluating SBC because the diagnostic is computationally prohibitive for non-amortized approaches.

Our results illustrate a clear distinction between the amortized and non-amortized paradigms. Amortized methods do not require retraining or new simulations to determine the empirical expected coverage probability of a posterior estimator, while non-amortized methods do. For this reason, a global coverage analysis of non-amortized approaches is computationally prohibitive and mostly impractical. More importantly, the coverage analysis of a non-amortized approach only measures the quality of the training procedure, whereas a coverage analysis of an amortized approach diagnoses the posterior estimator itself. In addition, a global coverage analysis not only serves as diagnostic but also allows to partially alleviate the issue by performing post-training calibration. A simple way for calibrating level $\alpha$ credible regions is to replace those by credible regions at a level that has the desired expected coverage. Finally, non-amortized sequential algorithms have to repeat the entire simulation-training pipeline whenever architectural or hyperparameter changes are made, while amortized methods reuse previously simulated datasets. All of the above lead us to conclude that while sequential methods have the benefit of being faster to train, amortized methods should be considered for sensitive applications requiring detailed statistical validation.

> **Observation 1** All benchmarked algorithms may produce non-conservative posterior approximations. This pathology tends to be accentuated with small simulation budgets in both paradigms.
>
> **Observation 2** Amortized approaches tend to be more conservative in contrast to non-amortized approaches.
>
> **Observation 3** The expected coverage probability of an ensemble model is larger than the average individual model. The ensemble size positively affects the expected coverage probability as well.
>
> **Observation 4** Amortized methods are simulation-efficient, especially when taking hyper-parameter tuning and the evaluation of the expected coverage diagnostic into account.

## 4 Discussion

As demonstrated empirically, simulation-based inference can be unreliable, especially whenever its approximations cannot be diagnosed. The problem of determining whether a posterior approximation is correct, or rather, suitable to a use case, is in fact not restricted to simulation-based inference specifically; the concern occurs in all of approximate Bayesian inference. The MCMC literature deals with this exact same problem in the form of determining whether a set of Markov chain samples have converged to the target distribution (Lin, 2014; Hogg & Foreman-Mackey, 2018). In this regard, empirical diagnostic tools have been proposed over the years (Geweke et al., 1991; Gelman & Rubin, 1992; Raftery & Lewis, 1991; Dixit & Roy, 2017; Talts et al., 2018) and have helped practitioners to apply MCMC reliably. While, we are not aware of any

coverage studies similar to ours for MCMC, we would expect similar results to be found. Nonetheless, there is currently no clear solution to determine convergence with absolute certainty (Dixit, 2018; Roy, 2020), even if the likelihood function is tractable. This issue has also been studied in the ABC literature. Prangle et al. (2014) present a way for approximating the coverage in ABC by reusing the same simulations in different runs. Xing et al. (2020) provides a diagnostic for the quality of the approximation for a given observation by the mean of distortion maps. Finally, Frazier et al. (2018) provide condition on the threshold $\epsilon$ such as to produce posterior estimates that asymptotically have proper frequentist coverage. However, the quality of the posterior estimates remains uncertain in a practical non-asymptotic regime.

We are of the opinion that theoretical and methodological advances within the field of simulation-based inference will strengthen its reliability and thereby promote its applicability in sciences. First, although all benchmarked algorithms recover the true posterior under specific optimal conditions, it is generally not possible to know whether those conditions are satisfied in practice. Therefore, the study of new objective functions that would force posterior estimators to always be conservative, regardless of optimality conditions, constitutes a valuable research avenue. From a Bayesian perspective, Rozet & Louppe (2021b) propose using the focal and the peripheral losses to weigh down strongly classified samples as a means to tune the conservativeness of a posterior estimator. However, the technique is empirical and requires tuning to attain the desired properties in practice. Dalmasso et al. (2020) on the other hand consider the frequentist setting and introduce a theoretically-grounded algorithm for the construction of confidence intervals that are guaranteed to have calibrated coverage, regardless of the quality of the used statistic. Dalmasso et al. (2021) extends this work with finite sample guarantees.

Second, in light of our results that ensembles produce more conservative posteriors, model averaging constitutes another promising direction of study as a simple and directly applicable method to produce reliable posterior estimators. However, a deeper understanding of the behaviour we observe is certainly first required to further develop these methods.

Third, post-training calibration can be used to improve the reliability of posterior estimators and should certainly be considered as a way toward more conservative inference. To some extent, this has already been considered for amortized methods (Cranmer et al., 2015; Brehmer et al., 2018; Hermans et al., 2021) and would be worth exploring further, especially for non-amortized approaches.

In this work, we assumed that the simulator perfectly models the real data generating process and focused on the computational faithfulness of the inference engine. In practice, there might be a gap between the simulation and reality. This issue, orthogonal to our work, should also be taken into account for making reliable simulation-based inference, as already initiated by several authors (Schmon et al., 2020; Matsubara et al., 2022; Pacchiardi & Dutta, 2022; Dellaporta et al., 2022).

In summary, we show that current algorithms for simulation-based inference can produce overconfident posterior approximations, making them possibly unreliable for scientific use cases and falsificationist inquiry. Nevertheless, we remain confident and optimistic and advocate that our results are only a stepping stone toward more reliable simulation-based inference and its wider adoption in the sciences.

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

# A    Estimation of the empirical expected coverage probability

We describe in this section the methodology used to compute the empirical expected coverage probability

$$\frac{1}{n} \sum_{i=1}^{n} \mathbb{1}\left(\boldsymbol{\theta}_i^* \in \Theta_{\hat{p}(\boldsymbol{\theta}|\boldsymbol{x}_i)}(1-\alpha)\right). \tag{7}$$

We consider $n$ test simulations $(\boldsymbol{\theta}_i^*, \boldsymbol{x}_i) \sim p(\boldsymbol{\theta})p(\boldsymbol{x}|\boldsymbol{\theta})$ and compute their associated approximate posteriors $\hat{p}(\boldsymbol{\theta}|\boldsymbol{x}_i)$ in a discretized and empirically normalized grid of the parameter space. The associated credible region is the highest density credible region, i.e. a credible region of the form

$$\Theta_{\hat{p}(\boldsymbol{\theta}|\boldsymbol{x}_i)}(1-\alpha) = \{\boldsymbol{\theta} : \hat{p}(\boldsymbol{\theta}|\boldsymbol{x}_i) \geq \gamma\}. \tag{8}$$

The threshold $\gamma$ is computed using a dichotomic search to produce a credible region of level $1 - \alpha$. We then estimate the empirical expected coverage probability by the proportion of nominal parameters $\boldsymbol{\theta}_i^*$ that falls in their associated credible region $\Theta_{\hat{p}(\boldsymbol{\theta}|\boldsymbol{x}_i)}(1-\alpha)$.

# B    Expected simulation times

| SLCP | M/G/1 | Weinberg | Lotka-V. | Spatial SIR | GW | Streams |
|---|---|---|---|---|---|---|
| $0.22 \pm 0.002$ | $0.20 \pm 0.002$ | $0.20 \pm 0.002$ | $19.08 \pm 0.96$ | $9.18 \pm 0.28$ | $545.13 \pm 23.63$ | $39,369 \pm 584$ |

Table 1: Expected simulation time to produce 1000 simulations for all benchmark problems on a single CPU core. The expected time and standard deviation are reported in seconds.

# C    Architectures & hyperparameters

In this section we describe the neural architectures and hyperparameters associated with our experiments. Our descriptions are complemented with the actual number of coverage evaluations. As evident from the tables describing both amortized and non-amortized approaches, the number of coverage evaluations for amortized approaches is substantially larger. It should be noted that, a coverage analysis consisting of 300 posteriors of the non-amortized approaches took months on these relatively simple problems. While for the amortized methods, a coverage analysis of 100,000 samples was a matter of hours to a few days depending on the dimensionality of $\boldsymbol{\theta}$.

## C.1    Amortized

### C.1.1    Neural Posterior Estimation

The MLP embeddings are 3 layer MLP's with 64 hidden units and a final latent space of 10, which is fed to the normalizing flow. The CNN architecture in the Gravitational Waves benchmark consists of a 13-layer deep convolutional head of 1D convolutions with a dilation factor of $2^d$. Where $d$ corresponds to the depth of the convolutional head. The SELU (Klambauer et al., 2017) function is used as an activation function.

### C.1.2    Neural Ratio Estimation

Our experiments use the ADAMW (Kingma & Ba, 2015; Loshchilov & Hutter, 2019) optimizer. Accross all benchmarks, the MLP architectures constitute of 3 hidden layers with 128 units and SELU (Klambauer et al., 2017) activations. The Gravitational Waves benchmark uses the same convolutional architecture as in NPE. The resulting embedding is flattened and fed to a MLP in which the dependence on the target parameter $\boldsymbol{\theta}$ is added. As before, the MLP consists of 3 hidden layers with 128 units.

|  | SLCP | M/G/1 | Weinberg | Lotka-V. | Spatial SIR | GW | Streams |
|---|---|---|---|---|---|---|---|
| *Embedding* | MLP | MLP | MLP | MLP | MLP | CNN | MLP |
| *Batch-size* | 128 | 128 | 128 | 128 | 128 | 64 | 128 |
| *Coverage samples individual* | 100,000 | 5,000 | 100,000 | 100,000 | 100,000 | 10,000 | 100,000 |
| *Coverage samples ensemble* | 20,000 | 5,000 | 20,000 | 20,000 | 20,000 | 5,000 | 20,000 |
| *Epochs* | 100 | 100 | 100 | 100 | 100 | 100 | 100 |
| *Model* | NSF | NSF | NSF | NSF | NSF | NSF | NSF |
| *Transforms* | 3 | 3 | 1 | 3 | 3 | 3 | 3 |
| *Learning-rate* | 0.001 | 0.001 | 0.001 | 0.001 | 0.001 | 0.001 | 0.001 |

Table 2: Architectures and hyperparameters associated with Neural Posterior Estimation.

|  | SLCP | M/G/1 | Weinberg | Lotka-V. | Spatial SIR | GW | Streams |
|---|---|---|---|---|---|---|---|
| *Architecture* | MLP | MLP | MLP | MLP | MLP | CNN | MLP |
| *Batch-size* | 128 | 128 | 128 | 128 | 128 | 64 | 128 |
| *Coverage samples individual* | 100,000 | 100,000 | 100,000 | 100,000 | 100,000 | 10,000 | 100,000 |
| *Coverage samples ensemble* | 20,000 | 20,000 | 20,000 | 20,000 | 20,000 | 10,000 | 20,000 |
| *Epochs* | 100 | 100 | 100 | 100 | 100 | 100 | 100 |
| *Learning-rate* | 0.001 | 0.001 | 0.001 | 0.001 | 0.001 | 0.001 | 0.001 |

Table 3: Architectures and hyperparameters associated with Neural Ratio Estimation.

## C.2 Non-amortized

All our implementations of non-amortized approaches rely on the reference implementation in `sbi` (Tejero-Cantero et al., 2020). We use the recommended defaults unless stated otherwise. Whenever available, the same MLP embedding network is used. It consists of 3 hidden layers with 64 units and SELU (Klambauer et al., 2017) activations. The latent space has a dimensionality of 10 features. For all sequential methods, we use 10 rounds to iteratively improve the posterior approximation. Tasks that are tagged with the **prohibitive** keyword are computationally prohibitive, but technically not intractable because the computational cost is tied to the learning of the posterior approximation and not to the underlying (intractable) likelihood model.

### C.2.1 SNPE

Our evaluations with SNPE specifically use the SNPE-C (Greenberg et al., 2019) variant, as suggested by `sbi` (Tejero-Cantero et al., 2020). To minimize inconsistencies between experiments, we use the defaults suggested by the `sbi` authors unless states otherwise. Specific changes are highlighted in Table 4.

|  | SLCP | M/G/1 | Weinberg | Lotka-V. | Spatial SIR | GW | Streams |
|---|---|---|---|---|---|---|---|
| *Batch-size* | 128 | 128 | 128 | 128 | 32 | Prohibitive | Prohibitive |
| *Coverage samples* | 300 | 300 | 300 | 300 | 300 | Prohibitive | Prohibitive |
| *Embedding* | MLP | MLP | MLP | MLP | MLP | Prohibitive | Prohibitive |
| *Epochs* | 100 | 100 | 100 | 100 | 100 | Prohibitive | Prohibitive |
| *Features* | 64 | 64 | 64 | 64 | 64 | Prohibitive | Prohibitive |
| *Model* | NSF | NSF | NSF | NSF | NSF | Prohibitive | Prohibitive |
| *Transforms* | 3 | 3 | 1 | 3 | 3 | Prohibitive | Prohibitive |
| *Rounds* | 10 | 10 | 10 | 10 | 10 | Prohibitive | Prohibitive |
| *Learning-rate* | 0.001 | 0.001 | 0.001 | 0.001 | 0.001 | Prohibitive | Prohibitive |

Table 4: Architectures and hyperparameters associated with Sequential Neural Posterior Estimation.

### C.2.2 SNL

In contrast to other sequential methods, our evaluations with SNL (Papamakarios et al., 2019) add two additional computationally prohibitive or Prohibitive benchmarks. At the root of this issue lies the dimensionality of the observable. In both cases, the dimensionality of observables caused memory issues in SNL. In addition, training a seperate embedding model (that requires additional simulations) is outside of the scope of this work. For this reason, we consider the Lotka-Volterra en Spatial SIR benchmark to be Prohibitive.

|  | SLCP | M/G/1 | Weinberg | Lotka-V. | Spatial SIR | GW | Streams |
|---|---|---|---|---|---|---|---|
| *Batch-size* | 128 | 128 | 128 | Prohibitive | Prohibitive | Prohibitive | Prohibitive |
| *Coverage samples* | 300 | 300 | 300 | Prohibitive | Prohibitive | Prohibitive | Prohibitive |
| *Epochs* | 100 | 100 | 100 | Prohibitive | Prohibitive | Prohibitive | Prohibitive |
| *Features* | 64 | 64 | 64 | Prohibitive | Prohibitive | Prohibitive | Prohibitive |
| *Model* | NSF | NSF | NSF | Prohibitive | Prohibitive | Prohibitive | Prohibitive |
| *Transforms* | 3 | 3 | 1 | Prohibitive | Prohibitive | Prohibitive | Prohibitive |
| *Rounds* | 10 | 10 | 10 | Prohibitive | Prohibitive | Prohibitive | Prohibitive |
| *Learning-rate* | 0.001 | 0.001 | 0.001 | Prohibitive | Prohibitive | Prohibitive | Prohibitive |

Table 5: Architectures and hyperparameters associated with Sequential Neural Likelihood.

### C.2.3 SNRE

|  | SLCP | M/G/1 | Weinberg | Lotka-V. | Spatial SIR | GW | Streams |
|---|---|---|---|---|---|---|---|
| *Architecture* | MLP | MLP | MLP | MLP | MLP | Prohibitive | Prohibitive |
| *Batch-size* | 128 | 128 | 128 | 128 | 128 | Prohibitive | Prohibitive |
| *Coverage samples* | 300 | 300 | 300 | 300 | 300 | Prohibitive | Prohibitive |
| *Epochs* | 100 | 100 | 100 | 100 | 100 | Prohibitive | Prohibitive |
| *Features* | 64 | 64 | 64 | 64 | 64 | Prohibitive | Prohibitive |
| *Rounds* | 10 | 10 | 10 | 10 | 10 | Prohibitive | Prohibitive |
| *Learning-rate* | 0.001 | 0.001 | 0.001 | 0.001 | 0.001 | Prohibitive | Prohibitive |

Table 6: Architectures and hyperparameters associated with Sequential Neural Ratio Estimation.

### C.2.4 Approximate Bayesian Computation

Our ABC implementation relies on the `MCABC` and `SMCABC` classes in the `sbi` (Tejero-Cantero et al., 2020) package. The specific settings from Rejection ABC and SMC-ABC are described in Tables 7 and 8 respectively. The quantile specifically refers to the proportion of closest samples that were kept in the final posterior. Because our specific implementation of coverage requires the ability to describe the posterior density function, we relied on Kernel Density Estimation to estimate the posterior density from the accepted samples.

|  | SLCP | M/G/1 | Weinberg | Lotka-V. | Spatial SIR | GW | Streams |
|---|---|---|---|---|---|---|---|
| *Coverage samples* | 300 | 300 | 300 | 300 | 300 | Prohibitive | Prohibitive |
| *Quantile* | 0.01 | 0.01 | 0.01 | 0.01 | 0.01 | Prohibitive | Prohibitive |

Table 7: Hyperparameters associated with Rejection Approximate Bayesian Computation.

|  | SLCP | M/G/1 | Weinberg | Lotka-V. | Spatial SIR | GW | Streams |
|---|---|---|---|---|---|---|---|
| *Coverage samples* | 300 | 300 | 300 | 300 | 300 | Prohibitive | Prohibitive |
| *$\epsilon$ decay* | 0.5 | 0.5 | 0.5 | 0.5 | 0.5 | Prohibitive | Prohibitive |
| *Quantile* | 0.01 | 0.01 | 0.01 | 0.01 | 0.01 | Prohibitive | Prohibitive |

Table 8: Hyperparameters associated with Sequential Monte Carlo Approximate Bayesian Computation.

## D    Predictive performance of the approximate posteriors

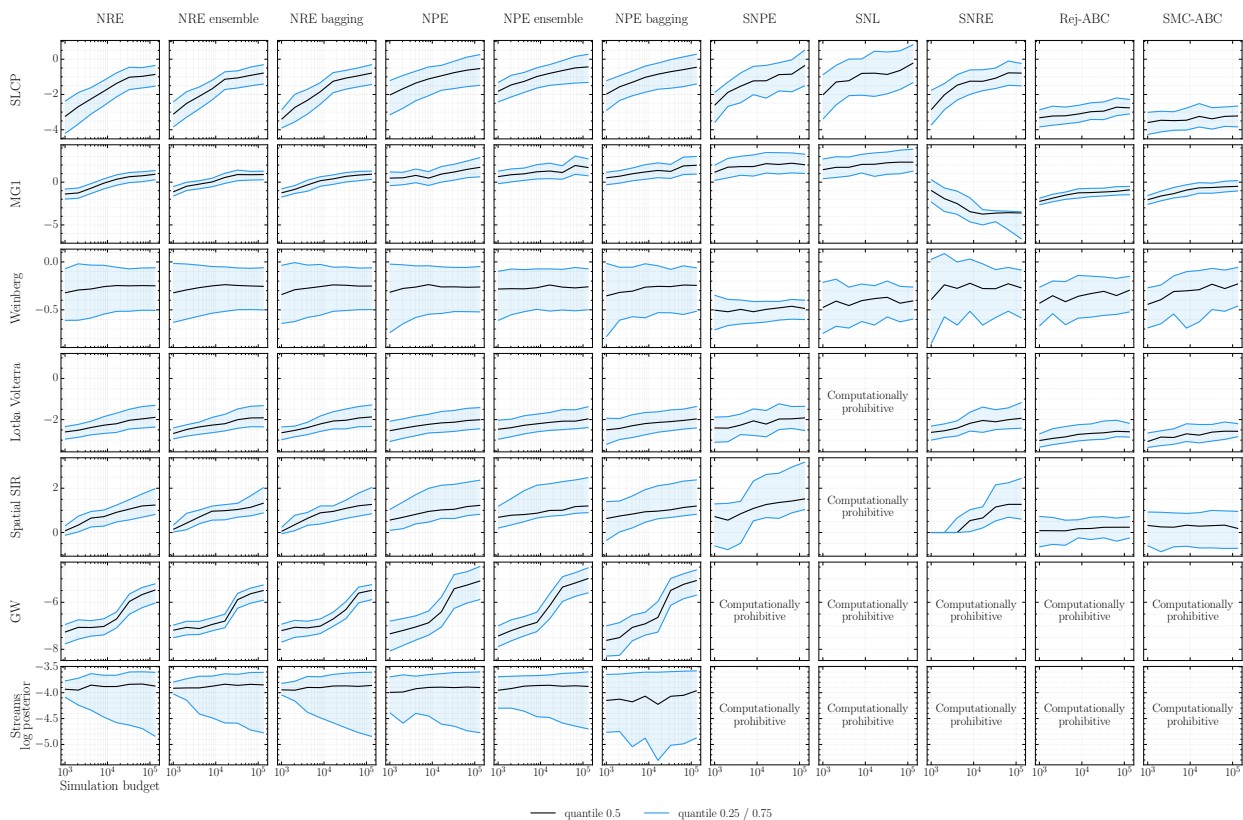

Figure 4: Evolution of the predicted log posterior probability of the nominal parameters with respect to the simulation budget. We evaluate the log posterior probability on $100,000$ test samples for amortized algorithms and $300$ for non-amortized algorithms. We report the $0.25; 0.5; 0.75$ quantiles of this quantity over the test set. We observe that the predictive performance increases in general with the simulation budget. Exceptions are the MG1 benchmark with the SNRE algorithm and the lowest quantile of the streams benchmark.

# E    Additional results

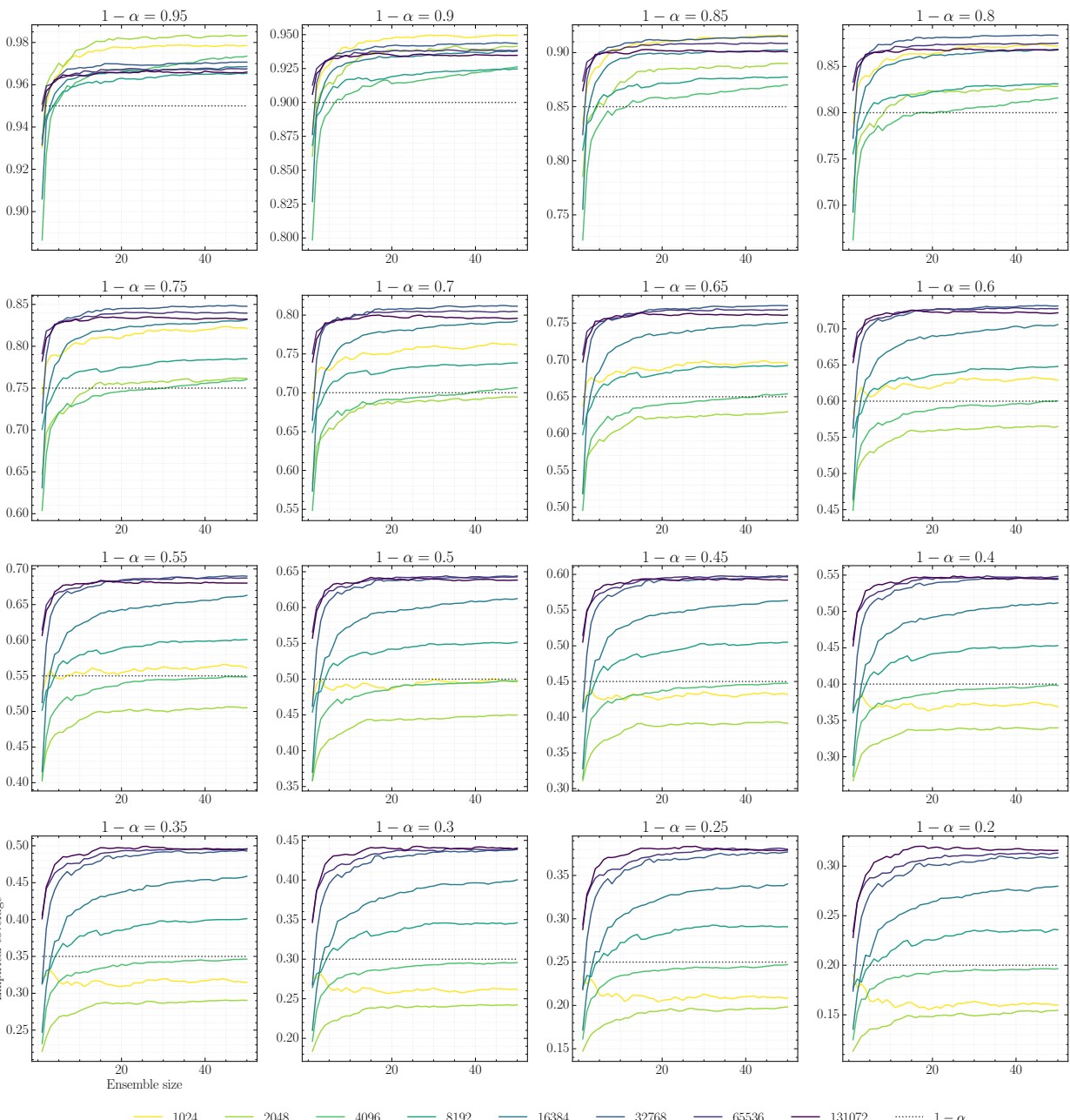

Figure 5: Evolution of the empirical expected coverage of deep ensembles with respect to ensemble size for various confidence levels. The results are obtained by training ratio estimators (NRE) on the SLCP benchmark. A positive effect is observed in terms of empirical expected coverage and ensemble size, i.e., a larger ensemble size correlates with a larger empirical expected coverage. This is unsurprising, because a larger ensemble is expected to capture more of the uncertainty that stems from the training procedure.

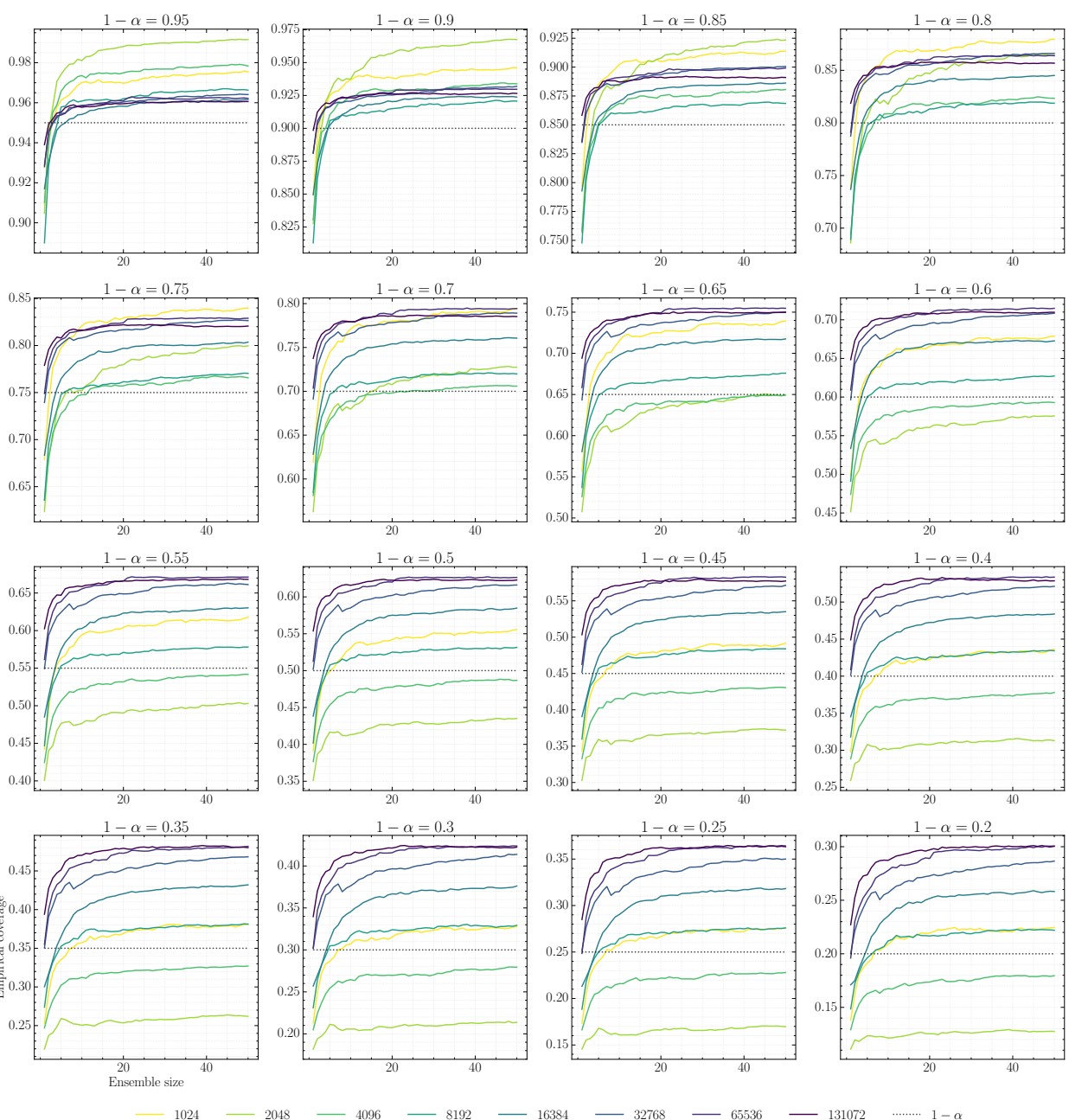

Figure 6: Evolution of the empirical expected coverage of bagging ensembles with respect to ensemble size for various confidence levels. The results are obtained by training ratio estimators (NRE) on the SLCP benchmark. A positive effect is observed in terms of empirical expected coverage and ensemble size, i.e., a larger ensemble size correlates with a larger empirical expected coverage. This is unsurprising, because a larger ensemble is expected to capture more of the uncertainty that stems from the training procedure.

