# OpenReview forum: "A Crisis In Simulation-Based Inference? Beware, Your Posterior Approximations Can Be Unfaithful"
_TMLR — Accepted by TMLR_

### Review · Reviewer_YGWM · 2022-09-19

**Summary Of Contributions:**

The paper provides empirical evaluation of a broad range of simulation-based inference methods from the perspective of whether they are overconfident or conservative. The authors propose a technically sound framework for evaluating this and then run a large collection of evaluations to provide an overall summary. In general, the methods are found to be overconfident, and this observation is lifted as the main conclusion of the work as an alert for the practitioners and methods developers. The authors also indicate that ensembles are better in this respect, and that simulation-efficient amortized methods perhaps have higher potential for sensitive applications.

**Broader Impact Concerns:**

The paper contributes by making limitations of core inference algorithms more transparent and hence has only positive broader impact; the results may help e.g. identifying scientific results or practices that are unreliable. This is communicated well throughout the paper and there is no need for addressing possible concerns in this respect.

**Requested Changes:**

The paper does not directly require any significant changes for publication (with the possible exception of the title), but I feel that addressing the following aspects would improve the paper by making the limitations and previous work more transparent and by making the value of the benchmark environment more visible for the community:

1. I feel that the title is too much of a click-bait and should be changed. It is a good idea to explicitly write "can be unfaithful" in the title to deliver the message directly, but I see no reason for calling it a 'crisis' or even using the word 'beware'. Your results do not show a surprising finding that undermines previous work, but rather just quantify better a behavior most people already expected and even show that many of the methods work reasonably well even though they are not perfect. Hence, calling the current situation 'crisis' is not justified.

2. You mention the parallel literature focusing on evaluation (of faithfulness) of standard inference algorithms only in Section 4. I would recommend mentioning this line of work for a highly related task already in Introduction. When I was reading the paper I was starting to get worried you are ignoring it completely. For example, Section 2.2 now implicitly suggests that looking at the posterior quality from the perspective of credible regions would be a novel idea, since you have not mentioned any previous approaches using the same principle in context of standard MCMC methods. You should avoid giving such impressions, especially when the value or your work does not depend at all on whether the specific measures are new or not. Finally, it would be good if you could more directly say whether you feel the issue of overconfident posterior estimates is more severe for simulation-based inference methods or whether practitioners should be aware of the same risks also for standard MCMC methods. This would naturally be done at the level of discussion, saying what kind of results others have found in past literature, not by actually running new experiments.

3. All of the benchmarks are low-dimensional; it seems the largest has 5 parameters. I understand this choice for a simulator-based inference paper that focuses on scientific validation of theoretical models, but it would still be useful to say something about models with larger parameter space. Would you expect the situation to get even worse if having 10+ parameters, or would it be better to indicate that this would need proper evaluation? Also, it would be good to indicate the scope of your empirical evaluation already in Introduction, explaining that you evaluate the methods in context of problems of specific dimensionality and explaining why this is the case.

4. One of your conclusions is that ensembling works well. I feel that you could emphasise the method a bit more already in the technical section 3.1.1. Now you have only a four-line summary of the idea, with no citations or motivation. If this is to be interpreted as a novel contribution of the work, I would change the writing here to explicitly say so and to provide a proper motivation and explanation for the method. If not, you should add references to previous works that have constructed such ensembles (in simulation-based inference or otherwise).

5. I see you provided code in a Supplement and the large file size indicates you must include some simulation results there as well, but the supplement is not very well documented and the manuscript does not state what is provided there. I recommend making this explicit. If you are making available both the simulation results (for which you spent significant amount of computation) and the code for running the evaluation metrics for new methods, then you are giving the community an extremely valuable resource since those enable evaluating faithfulness of future methods in a nice quantifiable benchmark. That would definitely be worth advertising better.


**Strengths And Weaknesses:**

**Strengths:**

1. The paper delivers an important message that can influence significant amount of methodological development and that provides practitioners useful empirical information. It is important to address the faithfulness of posterior approximations and methods papers should do this better.

2. The authors are very transparent in explaining how the empirical basis of the work required costly computation, which nicely justifies the need to publish the work to prevent others from needing to repeat similar effort to select a suitable algorithm for their specific case.

3. The paper is very well written and is a pleasure to read. Difficult concepts are nicely illustrated with examples, the somewhat complex inference algorithms are described with brief paragraphs that introduce the main idea without trying to unnecessarily explain the details here.

4. The technical development is sound; the chosen measures and the empirical setups are reasonable, the evaluated inference methods are a representative collection, and the benchmarks are diverse and interesting. I cannot pinpoint any specific weaknesses in the execution.

**Weaknesses:**

1. Too bold title.

2. Previous work on evaluating posterior quality from this perspective in context of standard inference algorithms could be better integrated into the story.

---

> ### Author Response · Authors · 2022-09-29
> **Rebuttal**
>
> Thank you for your positive review and for pointing out the importance, soundness and clarity of the manuscript.
>
> > I feel that the title is too much of a click-bait and should be changed. It is a good idea to explicitly write "can be unfaithful" in the title to deliver the message directly, but I see no reason for calling it a 'crisis' or even using the word 'beware'. Your results do not show a surprising finding that undermines previous work, but rather just quantify better a behavior most people already expected and even show that many of the methods work reasonably well even though they are not perfect. Hence, calling the current situation 'crisis' is not justified.
>
> We propose to change the title to `A Trust Crisis In Simulation-Based Inference? Your Posterior Approximations Can Be Unfaithful`. We believe that talking about a trust crisis softens the message. Even if this result was expected by researchers in the field, this can still lead to a trust crisis as the obtained posterior approximation could be unfaithful. In addition, we believe that this is not necessarily expected by practitioners.
>
> > You mention the parallel literature focusing on evaluation (of faithfulness) of standard inference algorithms only in Section 4. I would recommend mentioning this line of work for a highly related task already in Introduction. When I was reading the paper I was starting to get worried you are ignoring it completely. For example, Section 2.2 now implicitly suggests that looking at the posterior quality from the perspective of credible regions would be a novel idea, since you have not mentioned any previous approaches using the same principle in context of standard MCMC methods. You should avoid giving such impressions, especially when the value or your work does not depend at all on whether the specific measures are new or not. Finally, it would be good if you could more directly say whether you feel the issue of overconfident posterior estimates is more severe for simulation-based inference methods or whether practitioners should be aware of the same risks also for standard MCMC methods. This would naturally be done at the level of discussion, saying what kind of results others have found in past literature, not by actually running new experiments.
>
> Thank you for your advice regarding the presentation. We have now updated the paragraph preceding the definition of coverage to mention that this is not new and cited works performing coverage analyses. We have also added references to likelihood-based methods diagnostics in the introduction.
>
> Regarding the comparison to MCMC. We are not aware of any large-scale coverage analysis similar to ours. Would you have any pointers? MCMC methods could also be overconfident if the chain has not been run for long enough. However, giving a notion of how severe this issue is compared to simulation-based inference is difficult and probably dependent on the benchmark/hyper-parameters.
>
>
> > All of the benchmarks are low-dimensional; it seems the largest has 5 parameters. I understand this choice for a simulator-based inference paper that focuses on scientific validation of theoretical models, but it would still be useful to say something about models with larger parameter space. Would you expect the situation to get even worse if having 10+ parameters, or would it be better to indicate that this would need proper evaluation? Also, it would be good to indicate the scope of your empirical evaluation already in Introduction, explaining that you evaluate the methods in context of problems of specific dimensionality and explaining why this is the case.
>
> This is a valid point, we explain in the setup section that we only focus on low-dimensional problems because the coverage calculation becomes too expensive in higher dimensions. We believe that showing the issue in a low-dimensional setting is sufficient to demonstrate that the issue exists. To make it clearer we added the following sentence `Nevertheless, we expect the observed behaviours to be similar or accentuated in high-dimensional settings.`

---

> > ### Author Response · Authors · 2022-09-29
> > **Rebuttal**
> >
> > > One of your conclusions is that ensembling works well. I feel that you could emphasise the method a bit more already in the technical section 3.1.1. Now you have only a four-line summary of the idea, with no citations or motivation. If this is to be interpreted as a novel contribution of the work, I would change the writing here to explicitly say so and to provide a proper motivation and explanation for the method. If not, you should add references to previous works that have constructed such ensembles (in simulation-based inference or otherwise).
> >
> > We added a reference in which the ensembling method used is described and clarified the text. Regarding the contribution, it rather lies in the observation that ensembles lead to more conservative approximate posteriors than invidual models. This is why it is mainly discussed in the result section.
> >
> > > I see you provided code in a Supplement and the large file size indicates you must include some simulation results there as well, but the supplement is not very well documented and the manuscript does not state what is provided there. I recommend making this explicit. If you are making available both the simulation results (for which you spent significant amount of computation) and the code for running the evaluation metrics for new methods, then you are giving the community an extremely valuable resource since those enable evaluating faithfulness of future methods in a nice quantifiable benchmark. That would definitely be worth advertising better.
> >
> > The large file size is actually due to the fact that we included run notebooks containing the different results. Apart from that, only the code is provided as simulation results' size sums up to 300 GB. Those are however available on request for anyone who would need it.

---

> > ### Comment · Reviewer_YGWM · 2022-10-06
> > **Response to rebuttal**
> >
> > Thank you for the detailed responses. I read the revised version and think that it further improved from the original submission. The new title is better and suitable for the paper, and I think that the small additions and textual modifications are sufficient for addressing the past work in testing of MCMC algorithms.
> >
> > > Regarding the comparison to MCMC. We are not aware of any large-scale coverage analysis similar to ours. Would you have any pointers?
> >
> > Unfortunately not, so rather than relying on the observations of a single well-crafted study you would here need to briefly cover isolated studies directly. Even isolated studies seem to be rare, though, since the main efforts of quantifying overconfidence seem to relate to analysis of specific models and priors rather than inference algorithms. If you are not aware of attempts of such coverage analysis but feel that it would be useful also for standard MCMC, then I recommend you simply mention that.
> >
> > > This is a valid point, we explain in the setup section that we only focus on low-dimensional problems because the coverage calculation becomes too expensive in higher dimensions.
> >
> > This is highly useful clarification, because it makes the limitations of the approach more explicit. It is clearly better to say it directly rather than to leave the reader wondering about the reasons.
> >
> > > We added a reference in which the ensembling method used is described and clarified the text. Regarding the contribution, it rather lies in the observation that ensembles lead to more conservative approximate posteriors than invidual models. This is why it is mainly discussed in the result section.
> >
> > I understand the reasoning and agree that formally the description of the ensembling is sufficient, but I personally still would have added some motivational sentences in the description to guide the reader to pay attention to that part already in the first reading. You already mention the observation of ensembles resulting in more conservative approximation as one contribution of your work in Introduction, and hence can safely refer to the future empirical result also in the methods section to remind that this is where you introduce the method that turns out to be useful. For instance, by simply stating that you will later empirically find ensembles a practical way to provide more conservative approximations (but with the natural disadvantage of needing to run n different algorithms) and that this insight is a novel finding of your work even though the technical method of constructing the ensembles is not new.

---

> > > ### Author Response · Authors · 2022-10-07
> > > **Thanks for the suggestions**
> > >
> > > > Unfortunately not, so rather than relying on the observations of a single well-crafted study you would here need to briefly cover isolated studies directly. Even isolated studies seem to be rare, though, since the main efforts of quantifying overconfidence seem to relate to analysis of specific models and priors rather than inference algorithms. If you are not aware of attempts of such coverage analysis but feel that it would be useful also for standard MCMC, then I recommend you simply mention that.
> > >
> > >
> > > > I understand the reasoning and agree that formally the description of the ensembling is sufficient, but I personally still would have added some motivational sentences in the description to guide the reader to pay attention to that part already in the first reading. You already mention the observation of ensembles resulting in more conservative approximation as one contribution of your work in Introduction, and hence can safely refer to the future empirical result also in the methods section to remind that this is where you introduce the method that turns out to be useful. For instance, by simply stating that you will later empirically find ensembles a practical way to provide more conservative approximations (but with the natural disadvantage of needing to run n different algorithms) and that this insight is a novel finding of your work even though the technical method of constructing the ensembles is not new.
> > >
> > > Thanks for the valuable suggestions, this indeed improves the paper! We have updated the paragraph on MCMC in the discussion and the paragraph on ensembles in the method section.

---

### Review · Reviewer_zzma · 2022-09-20

**Summary Of Contributions:**

This paper studies empirically the coverage of credible intervals from Bayesian simulation-based approximate inference algorithms. Providing a sufficiently wide or conservative credible interval is more useful for scientific use cases than an over-confident estimator because the risk of excluding the truth value of the underlying parameter is much higher than including providing an conservative range. Based on this observation, the authors conduct empirical studies to estimate the over-confidence of recent simulation-based Bayesian inference methods on 5 synthetic and 2 real scientific problems. They observe that (1) all estimators can be over-confident for some problem and (2) amortised methods tend to provide more conservative estimators than non-amortised estimators (3) the empirical coverage of ensemble models is larger than individual models and (4) amortised methods are more simulation-efficient than non-amortised methods.

This paper provides an important perspective to evaluate the conservativeness of Bayesian inference methods for scientific discoveries that is orthogonal to the commonly used diagnostic metrics  such as the divergence of the posterior distribution. I think this makes a nice contribution to raise the awareness of practitioners for risks in applying ML estimators in this application domain. The empirical observation provides a good step towards developing conservative estimators with guarantees.


**Requested Changes:**

Listed above in the weaknesses.

**Strengths And Weaknesses:**

Strengths:
- This work is well motivated and presented. It’s easy to read and the conclusions are clearly presented and discussed. The details of experiment setting and code are provide, which is good for reproduction.
- The summarised osbervations are well supported by the experiments. The authors do not over-claim their contributions.
- The studied empirical coverage metric could be a useful tool to assess Bayesian estimators in additional to transitional metrics.

Weaknesses:
- The risk of excluding high probable regions of the parameter in inference is not a new problem. The proposed empirical coverage probably is the expected frequentist coverage probability as discussed by the authors below Equation 3. I would expect the authors to discuss how proposed method is related to and contribute to existing work along this ling of study in the literature later in the paper but didn’t find any.
- While the over-confidence is different from accuracy to assess an estimator, they’re still correlated. As an estimator becomes more conservative, the size of the credible interval and the divergence of its posterior from the true distribution will become larger. I’m worried that comparing esitmators wrt coverage probability only may give a delusion especially when the credible interval can be adjusted post-hoc by a different confidence level. Providing other metrics, e.g., size of credible region / AUROC / divergence along with coverage seems important as it provide a more comprehensive look in the main figures.
- In page 9, the authors discussed that using \epsilon > 0 doesn’t always correspond to conservative approximations. Can they show any experiments to support it? Why not present the best found conservative estimators but using the naive identity function? I guess there is tradeoff between conservativeness and accuracy for \epsilon, but that tradeoff exists when comparing other estimators too.

There are also a few minor concerns/comments
- How are the hyper-parameters of each method chosen? Is the coverage sensitive to the choice of hyper-parameters?
- How is ensemble bagging different from ensemble? Some more explanation would be appreciated.
- It may be useful to make it clear when talking about the main conclusion/observations that the empirical studies are restricted to low dimensional parameter spaces. The conclusion may not extend to high dimensional spaces.

---

> ### Author Response · Authors · 2022-09-29
> **Rebuttal**
>
> Thank you for your positive review and for pointing out the clarity of the manuscript and the soundness of the experiments.
>
> > The risk of excluding high probable regions of the parameter in inference is not a new problem. The proposed empirical coverage probably is the expected frequentist coverage probability as discussed by the authors below Equation 3. I would expect the authors to discuss how proposed method is related to and contribute to existing work along this ling of study in the literature later in the paper but didn’t find any.
>
> We have updated the paragraph preceding the definition of coverage to mention that this is not new and cited works performing coverage analyses. If there are any references that we would have missed, please inform us and we will add those.
>
>
> > While the over-confidence is different from accuracy to assess an estimator, they’re still correlated. As an estimator becomes more conservative, the size of the credible interval and the divergence of its posterior from the true distribution will become larger. I’m worried that comparing esitmators wrt coverage probability only may give a delusion especially when the credible interval can be adjusted post-hoc by a different confidence level. Providing other metrics, e.g., size of credible region / AUROC / divergence along with coverage seems important as it provide a more comprehensive look in the main figures.
>
> We agree and this is something we actually mention at the top of page 4: `For this reason, a complete analysis should be complemented with measures such as
> the mutual information or expected information gain`. However, this study focuses on the conservative aspect which we advocate is required no matter the accuracy of the approximate posterior. For this reason, we made the choice of only computing this metric to emphasize this aspect. That being said, we understand this a valuable piece of information for a more nuanced discussion. We are currently running scripts over our checkpointed models to evaluate one such metric and will update the manuscript to include it in the next few days.
>
> > In page 9, the authors discussed that using $\epsilon > 0$ doesn’t always correspond to conservative approximations. Can they show any experiments to support it? Why not present the best found conservative estimators but using the naive identity function? I guess there is tradeoff between conservativeness and accuracy for $\epsilon$, but that tradeoff exists when comparing other estimators too.
>
> In the limit $\epsilon = 0$, you accept $\theta$ values that generated synthetic observations identical to the real observation and hence recover the true posterior $p(\theta|x = x_o)$. When you increase $\epsilon$ you accept $\theta$ values that generated synthetic observations further from the real observation. The way it modifies the approximate posterior is unknown, it could increase the variance but also shift the distribution and hence there are no guarantees regarding the coverage. Experimentally, this is shown in Figure 2. The experiments have been performed with $\epsilon > 0$ and it sometimes leads to non-conservative posterior approximations.
>
> We made the choice of using a naive identity function to accentuate the fact that even without compressing the observation into a summary statistic, the resulting approximation can be non-conservative. Of course, when targetting high accuracy, using a summary statistic is preferred.
>
> There is indeed a tradeoff in setting the $\epsilon$ value but this is more of an efficiency/accuracy tradeoff. When increasing $\epsilon$ you lose accuracy as you accept samples far from the real observation. When decreasing $\epsilon$ you lose efficiency as a higher proportion of samples gets rejected and hence more simulations are needed to get the same amount of samples.

---

> > ### Comment · Reviewer_zzma · 2022-10-03
> > **Some questions remain**
> >
> > Thanks for providing the clarification. Some of my questions such as the use of $\epsilon$ in ABC have been addressed by the feedback (please provide those hyperparameters in the appendix). I would expect the authors to make more edits in their revision for my other comments.
> >
> > I share the same concern as reviewer dket that another metric such as the size of the credible region should be provided along with the proposed coverage probability, which the authors promise to provide in a few days.
> >
> > I would hope the authors could explain more on how the proposed method contributes to / differs from the existing literature the authors added in the revision, how the ensembe (bagging) method differs from the deep ensembling method. The authors added in the revision "Nevertheless, we expect the observed behaviours to be similar or accentuated in a high-dimensional setting." Is there any justification for it?

---

> > > ### Author Response · Authors · 2022-10-07
> > > **Further clarification**
> > >
> > > > Thanks for providing the clarification. Some of my questions such as the use of $\epsilon$ in ABC have been addressed by the feedback (please provide those hyperparameters in the appendix). I would expect the authors to make more edits in their revision for my other comments.
> > >
> > > We have updated the manuscript again to include the $\epsilon$ value. The following sentence has also been updated: `In our experiments, we use the identity function as a sufficient summary statistic and set $\epsilon$ such that $\max(100, \text{simulation budget} / 100)$ samples are accepted.`
> > >
> > > Regarding the other comments:
> > > * We had already updated the paragraph preceding the definition of coverage to mention that this is not new and we have now added citations of works performing coverage analyses.
> > > * We have measured the performance of all the algorithms and have added a figure summarizing the results.
> > > * We have clarified that our conclusions hold for low-dimensional parameter spaces.
> > > * We have clarified how ensemble bagging is different from ensemble (see below).
> > > * Regarding the choice of the hyper-parameters. Those have been chosen such as to make training relatively fast as there are a lot of models to train but still provide acceptable performance.
> > >
> > > If some uncertainty remains, please let us know precisely which piece of information is missing for us to edit the manuscript.
> > >
> > > > I share the same concern as reviewer dket that another metric such as the size of the credible region should be provided along with the proposed coverage probability, which the authors promise to provide in a few days.
> > >
> > > This is now done in the latest revision (Fig 4). We used the log posterior of the nominal parameter value as a performance metric. We conclude from this figure that on most benchmark/method pairs, the predictive performance increases with the simulation budget. However, this is not the case for SNRE applied to MG1 which explains the strange behaviour of the coverage curves.
> > >
> > > > I would hope the authors could explain more on how the proposed method contributes to / differs from the existing literature the authors added in the revision, how the ensembe (bagging) method differs from the deep ensembling method. The authors added in the revision "Nevertheless, we expect the observed behaviours to be similar or accentuated in a high-dimensional setting." Is there any justification for it?
> > >
> > > First, we would like to emphasize that we do not frame our contribution as a new ensembling method but rather as the empirical demonstration that well-established and common ensembling techniques provide more conservative results than using a single model. We have clarified the difference between the two in the ensemble paragraph and provided references for both methods. `In this work, we
> > > consider an ensemble model that averages the approximated posteriors of n posterior estimators independently trained on the same dataset Lakshminarayanan et al. (2017) or on bootstrap replicates of the learning set (bagging) Breiman (1996).`
> > >
> > >
> > > Regarding the high-dimensional setting, the justification would be that we do not see any reason why it would not be the case and increasing the parameter dimensionality would solve the issue.
> > >
> > > *Leo Breiman. Bagging predictors. Machine learning, 24(2):123–140, 1996.*
> > >
> > > *Balaji Lakshminarayanan, Alexander Pritzel, and Charles Blundell. Simple and scalable predictive uncertainty estimation using deep ensembles. Advances in neural information processing systems, 30, 2017.*

---

### Review · Reviewer_dket · 2022-09-23

**Summary Of Contributions:**

This paper provides an empirical investigation of a variety of simulation-based (also called “likelihood-free”) statistical inference algorithms, which are used when (a) the mode likelihood cannot be computed but (b) data can be simulated from the model given the model parameters. Simulation-based inference methods are used in a range of scientific applications ranging from physics to ecology to population genetics. The paper is particularly concerned with the calibration properties of simulation-based algorithms: are the credible regions of the algorithms well-calibrated when the true parameter is drawn from the prior? Using a range of toy and realistic models, the authors show that all of the methods considered could significantly underestimate parameter uncertainty. While non-amortized methods were cheaper to run for a single problem, the authors argue that amortized methods are easier to validate in the kind of simulation-based framework they develop, where many datasets are generated from the model conditional on parameters sampled from the prior.

**Broader Impact Concerns:**

I have no concerns

**Requested Changes:**

1. Include comparisons to a more modern version ABC and synthetic likelihood
2. Discuss how the results relate to existing theory for ABC and synthetic likelihood methods, and correct the claim that “advances in the field [of simulation-based inference] are mainly driven from a machine learning perspective.”
3. Include results on the volume of the credible regions for each method.
4. Clarify the ensembling method used and its connection to existing Bayesian flavored bootstrapping/bagging methods.
5. Address all minor issues


**Strengths And Weaknesses:**

Overall I found the paper to be well-written. Most of the claims made by the authors are supported by the experiments. However, there are a few weaknesses that need to be addressed.

1. The paper does not adequately consider – both in the discussions and the experiments – the significant amount of activity in the statistics literature on likelihood-free methods. In particular, there have been substantial methodological and theoretical advances in ABC and synthetic likelihood. For example, here is a short, very incomplete list:

* Frazier, D. T., Nott, D. J., Drovandi, C. & Kohn, R. Bayesian Inference Using Synthetic Likelihood: Asymptotics and Adjustments. J Am Stat Assoc (2022).
* Frazier, D. T., Robert, C. P. & Rousseau, J. Model misspecification in approximate Bayesian computation: consequences and diagnostics. Journal of the Royal Statistical Society: Series B (Statistical Methodology) 162, 2025–24 (2020).
* Li, W. & Fearnhead, P. On the asymptotic efficiency of approximate Bayesian computation estimators. Biometrika 105, 285–299 (2018).
* ​​Karabatsos, G. & Leisen, F. An approximate likelihood perspective on ABC methods. Statistics Surveys 12, 66–104 (2018).
* Frazier, D. T., Martin, G. M., Robert, C. P. & Rousseau, J. Asymptotic properties of approximate Bayesian computation. Biometrika 105, 593–607 (2018).
* Moral, P. D., Doucet, A. & Jasra, A. An adaptive sequential Monte Carlo method for approximate Bayesian computation. Statistics and Computing 22, 1009–1020 (2012).
* Fearnhead, P. & Prangle, D. Constructing summary statistics for approximate Bayesian computation: semi‐automatic approximate Bayesian computation. 74, 419–474 (2012).
* Blum, M. G. B. Approximate Bayesian computation: A nonparametric perspective. Journal of the American Statistical Association 105, (2010).

The basic ABC method used in the comparisons is known to be very problematic. Far better alternatives such as MCMC-ABC and SMC-ABC are available. Moreover, techniques like regression adjustment can substantially improve inferences.

2. As the authors point out at the top of p. 4, in addition to having well-calibrated credible sets, it is also crucial to consider the size of such sets to determine if any meaningful inferences are being drawn. The authors try to sidestep this question by placing it out of scope of their paper. However, I do not think this can be done: to make the comparisons meaningful, something must be said about whether these methods are, in fact, learning anything from the data! While attempting to estimate something like the expected information gain presents significant challenges, there are simpler metrics that would provide substantial insights. For example, using the results of the existing experiments, it is trivial to compute the volume of the credible regions of each method.

3. The ensemble method used is not clearly described, to the degree that I’m unsure exactly what was done. It seems that the proposed ensembling technique may be very similar to BayesBag or various other bootstrapping methods:

* Pompe, E. Introducing prior information in Weighted Likelihood Bootstrap with applications to model misspecification. arxiv.org arXiv:2103.14445v2 [stat.ME] (2021).
* Huggins, J. H. & Miller, J. W. Robust Inference and Model Criticism Using Bagged Posteriors. arxiv.org arXiv:1912.07104 [stat.ME] (2019).
* Lyddon, S. P., Holmes, C. C. & Walker, S. G. General Bayesian updating and the loss-likelihood bootstrap. Biometrika 106, 465–478 (2019).
* Lyddon, S. P., Walker, S. G. & Holmes, C. C. Nonparametric learning from Bayesian models with randomized objective functions. In 32nd Conference on Neural Information Processing Systems (2018).

4. There are a few additional minor issues:

* The abstract used numerous undefined acronyms
* p. 1: “they are unfortunately not suitable for statistical inference​​.” This is false. They make statistical inference more challenging.
* In the introduction it would be nice to list some applications outside of physics.
* Matsubara et al. has been published.

---

> ### Author Response · Authors · 2022-09-29
> **Rebuttal**
>
> > The paper does not adequately consider – both in the discussions and the experiments – the significant amount of activity in the statistics literature on likelihood-free methods. In particular, there have been substantial methodological and theoretical advances in ABC and synthetic likelihood. For example, here is a short, very incomplete list:
>
> >Frazier, D. T., Nott, D. J., Drovandi, C. & Kohn, R. Bayesian Inference Using Synthetic Likelihood: Asymptotics and Adjustments. J Am Stat Assoc (2022).
> Frazier, D. T., Robert, C. P. & Rousseau, J. Model misspecification in approximate Bayesian computation: consequences and diagnostics. Journal of the Royal Statistical Society: Series B (Statistical Methodology) 162, 2025–24 (2020).
> Li, W. & Fearnhead, P. On the asymptotic efficiency of approximate Bayesian computation estimators. Biometrika 105, 285–299 (2018).
> ​​Karabatsos, G. & Leisen, F. An approximate likelihood perspective on ABC methods. Statistics Surveys 12, 66–104 (2018).
> Frazier, D. T., Martin, G. M., Robert, C. P. & Rousseau, J. Asymptotic properties of approximate Bayesian computation. Biometrika 105, 593–607 (2018).
> Moral, P. D., Doucet, A. & Jasra, A. An adaptive sequential Monte Carlo method for approximate Bayesian computation. Statistics and Computing 22, 1009–1020 (2012).
> Fearnhead, P. & Prangle, D. Constructing summary statistics for approximate Bayesian computation: semi‐automatic approximate Bayesian computation. 74, 419–474 (2012).
> Blum, M. G. B. Approximate Bayesian computation: A nonparametric perspective. Journal of the American Statistical Association 105, (2010).
>
> Thanks for the references. We are aware that there exists a wide literature on ABC and our goal is not to undermine it. In this paper, we focus specifically on the overconfidence issue in a practical setting and hence discuss papers that are directly related to this issue. From what we understand, the papers you sent focus on:
>
> * ABC statistical performance
> * ABC reliability in asymptotic regimes
> * ABC reliability under model misspecification
>
> Those are important topics but orthogonal to our work. In this regard, we already discuss [1] but did not find other related works. However, we are less familiar with the ABC literature than with the neural-SBI one and would be glad to discuss any paper relevant to our work that we would have missed.
>
> [1] Dennis Prangle, Michael GB Blum, G Popovic, and SA Sisson. Diagnostic tools for approximate bayesian computation using the coverage property. Australian & New Zealand Journal of Statistics, 56(4):309–329, 2014.
>
> > As the authors point out at the top of p. 4, in addition to having well-calibrated credible sets, it is also crucial to consider the size of such sets to determine if any meaningful inferences are being drawn. The authors try to sidestep this question by placing it out of scope of their paper. However, I do not think this can be done: to make the comparisons meaningful, something must be said about whether these methods are, in fact, learning anything from the data! While attempting to estimate something like the expected information gain presents significant challenges, there are simpler metrics that would provide substantial insights. For example, using the results of the existing experiments, it is trivial to compute the volume of the credible regions of each method.
>
> We believe that our results regarding coverage are valid no matter the performance of the algorithm. Indeed, in the ultimate case where nothing can be learned from the data, one still hopes that the model will be conservative and recover the prior instead of making overconfident erroneous inferences. Our goal is not to make a comparison between the different algorithms (this has already been done in [2]) but rather to showcase an issue shared by all existing algorithms. That being said, we understand this a valuable piece of information for a more nuanced discussion. We are currently running scripts over our checkpointed models to evaluate an additional metric and will update the manuscript to include those results in the next few days.
>
> [2] Lueckmann, J. M., Boelts, J., Greenberg, D., Goncalves, P., & Macke, J. (2021, March). Benchmarking simulation-based inference. In International Conference on Artificial Intelligence and Statistics (pp. 343-351). PMLR.
>
> >The basic ABC method used in the comparisons is known to be very problematic. Far better alternatives such as MCMC-ABC and SMC-ABC are available. Moreover, techniques like regression adjustment can substantially improve inferences.
>
> SMC-ABC is actually included in our experiments. The reason why we made the choice of using the simple rej-ABC method is that again we do not aim to compare the different methods but to showcase the overconfidence issue. In this regard, using the simplest version removes all the artifacts and demonstrates that this method has this issue even in its simplest form.

---

> > ### Author Response · Authors · 2022-09-29
> > **Rebuttal**
> >
> > > The ensemble method used is not clearly described, to the degree that I’m unsure exactly what was done. It seems that the proposed ensembling technique may be very similar to BayesBag or various other bootstrapping methods:
> >
> > > Pompe, E. Introducing prior information in Weighted Likelihood Bootstrap with applications to model misspecification. arxiv.org arXiv:2103.14445v2 [stat.ME] (2021).
> > Huggins, J. H. & Miller, J. W. Robust Inference and Model Criticism Using Bagged Posteriors. arxiv.org arXiv:1912.07104 [stat.ME] (2019).
> > Lyddon, S. P., Holmes, C. C. & Walker, S. G. General Bayesian updating and the loss-likelihood bootstrap. Biometrika 106, 465–478 (2019).
> > Lyddon, S. P., Walker, S. G. & Holmes, C. C. Nonparametric learning from Bayesian models with randomized objective functions. In 32nd Conference on Neural Information Processing Systems (2018).
> >
> > We have updated the manuscript to clarify the ensembling method used. It is actually quite simple, we train several members on the same dataset and the randomness comes from the training procedure.
> >
> > > There are a few additional minor issues:
> > The abstract used numerous undefined acronyms
> > p. 1: “they are unfortunately not suitable for statistical inference​​.” This is false. They make statistical inference more challenging.
> > In the introduction it would be nice to list some applications outside of physics.
> > Matsubara et al. has been published.
> >
> > We have addressed all the issues except the acronyms in the abstract. Those acronyms are well known in the sbi community and are defined in the paper. We believe that defining those in the abstract would make it unnecessarily heavy.

---

> > > ### Comment · Reviewer_dket · 2022-10-06
> > > **Reply to rebuttal**
> > >
> > > Thank you for the detailed response and clarifications, and the updates to the manuscript. In addition the credible interval sizes (which I know you are working), there are a few issues that still need to be addressed:
> > >
> > > 1. Regarding the references I provided, these were meant to illustrate the large and active literature rather than be a list of papers you necessarily need to cite or engage with. However, I would expect you to make a reasonable effort to be up-to-date with the ABC and synthetic likelihood literature and cite/discussion what is appropriate, given the importance of these methods. For example, how do your results relate to the existing asymptotic theory for ABC?
> > >
> > > 2. Regarding the ABC comparisons, my apologies for overlooking the inclusion of SMC-ABC. That said, I would still like to see ABC with regression adjustment, as that is a more state-of-the-art comparison than vanilla ABC. For completeness, you should also include a comparison to synthetic likelihood.
> > >
> > > 3. While you may be familiar with the acronyms, it is inappropriate to assume your reader will be. Moreover, acronyms are or could become overloaded. So, you should never introduce one without writing it out first (with a small number of exceptions; e.g., DNA). They need to be written out in the abstract.
> > >
> > > Finally, although this is certainly not necessary for acceptance, I think it would be really interesting to include a comparison to the “bagging/bootstrapping” version of the ensembling method, as this could really help with calibration even though the models are well-specified.

---

> > > > ### Author Response · Authors · 2022-10-08
> > > > **More updates have been made**
> > > >
> > > > > Regarding the references I provided, these were meant to illustrate the large and active literature rather than be a list of papers you necessarily need to cite or engage with. However, I would expect you to make a reasonable effort to be up-to-date with the ABC and synthetic likelihood literature and cite/discussion what is appropriate, given the importance of these methods. For example, how do your results relate to the existing asymptotic theory for ABC?
> > > >
> > > > We have now updated the method section to include regression adjustment, MCMC-ABC, automatic summary statistics learning and synthetic likelihoods. We have also updated the discussion to include the asymptotic theory for ABC.
> > > >
> > > > > Regarding the ABC comparisons, my apologies for overlooking the inclusion of SMC-ABC. That said, I would still like to see ABC with regression adjustment, as that is a more state-of-the-art comparison than vanilla ABC. For completeness, you should also include a comparison to synthetic likelihood.
> > > >
> > > > We agree that those improvements would certainly increase the predictive performance and have mentionned those in the text (see above). The choice of the algorithms we use is mainly based on what is available in the [sbi library](https://www.mackelab.org/sbi/). The rationale is that practitioners would pick one of these with default hyperparameter values and standard neural architectures. From this point of view the aim of the paper is to study the overconfidence issue in a practical setting. We could have extended the study to other algorithms and more evolved neural network architectures but it would come with an increased computational cost which is already high.
> > > >
> > > > > Finally, although this is certainly not necessary for acceptance, I think it would be really interesting to include a comparison to the “bagging/bootstrapping” version of the ensembling method, as this could really help with calibration even though the models are well-specified.
> > > >
> > > > We believe that we already compare the two methodologies in Figure 3. If you are mentionning another method, could you provide a reference to it?

---

### Author Response · Authors · 2022-09-29
**General comment**

First and foremost we would like to thank the reviewers for the high quality of their reviews and the positive reception of our work regarding the importance of the message, the quality of the writing, and the soundness of the experiments. We appreciate the suggestions and all the missing experiments will be submitted by the end of the discussion period.

---

> ### Author Response · Authors · 2022-10-07
> **Experiments have been updated**
>
> The experiments have been updated in the latest revision (Fig 4). We used the log posterior of the nominal parameter value as a performance metric. We conclude from this figure that on most benchmark/method pairs, the predictive performance increases with the simulation budget. However, this is not the case for SNRE applied to MG1 which explains the strange behaviour of the coverage curves.

---

### Decision · Action_Editors · 2022-10-28

**Recommendation:** Accept with minor revision

**Comment:**

The authors have already made many of the changes requested by the reviewers. I thank both the reviewers and the authors for their fruitful conversation during the review period.

I am following up on one reviewer who wishes to see a few minor changes:
- define the acronyms in the abstract
- update the final paragraph of section 2 to reflect the additional predictive accuracy experiments
- include results for both the deep ensemble and bagging in figures 4 and 5

With that I am happy to accept this manuscript and look forward to seeing it published.


**Audience:**

The area of simulation-based inference is growing. I believe there will certainly be interest in TMLR's audience for this comparative study.

**Claims And Evidence:**

All three reviewers have found the empirical study presented in this manuscript to be accurate and clear. They have raised a variety of suggestions that the authors have incorporated in their revision. The reviewers provided excellent feedback, and the discussion with the authors was informative and fruitful.

---

> ### Author Response · Authors · 2022-11-02
> **Thanks**
>
> Many thanks to the reviewers and action editor for the time spent reviewing our paper and the great discussions! We are currently running experiments to include bagging in figures 4 and 5 and will upload the camera-ready version once we get the results.

---

> ### Author Response · Authors · 2022-11-15
> **Camera-ready**
>
> Dear reviewers and action editor,
> We have now uploaded a camera-ready revision with all the requested changes.